# From Ambiguous Feedback to Verifiable Repair via Formal Synthesis in Text-to-SQL

## Abstract

Large Language Models (LLMs) for Text-to-SQL frequently generate formally incorrect queries, yet existing repair methods rely on diagnostically impoverished execution feedback. This forces LLMs to speculate on error causes, undermining reliability. We introduce Post-SQLFix, a neuro-symbolic framework instigates a paradigm shift from ambiguous feedback to verifiable repair, designed to supersede the database execute feedback paradigm. Our approach canonicalizes any SQL query into a dialect-aware, canonical query structure (CQS) representation. Upon CQS, a symbolic engine performs systematic syntactic, context-free, and context-sensitive semantic analysis to produce a sound diagnosis. We then formalize repair as a constrained synthesis problem: for any detected violation, our engine synthesizes a constrained space of formally verifiable repair plans. This transforms the LLM from an unreliable corrector into a constrained agent tasked with implementing a valid, synthesized plan. On the BIRD and Spider benchmarks, including multi-dialect subsets, Post-SQLFix boosts execution accuracy by up to +11.6% and reduces repair iterations by 50% compared to execution feedback. By replacing ambiguous feedback with formal guarantees, our framework represents a significant step towards building robust and trustworthy AI-driven code generation.

## 1 Introduction

Large Language Models (LLMs) have achieved strong performance in Text-to-SQL translation (Yu et al., 2018; Wang et al., 2020; Hwang et al., 2019), demonstrating the ability to map natural-language queries to executable SQL. Nevertheless, LLMs remain brittle: they frequently contain syntactic violations, schema-entity and type mismatches, or logical errors (e.g., incorrect joins, predicates, or aggregation scopes) (Yu et al., 2018; Wang et al., 2020).

To mitigate these reliability issues, many systems adopt an **LLM-as-a-judge** self-correction pipeline, where the model repairs its own SQL (Pourreza & Rafiei, 2023; Gao et al., 2024). Yet the same probabilistic model that produced the error is asked to diagnose and fix it (Madaan et al., 2023); studies show such self-correction can stall or even degrade reasoning (Huang et al., 2024; Kamoi et al., 2024), underscoring the need for external, trustworthy signals. A complementary thread adds external knowledge via retrieval or multi-agent designs. Retrieval-augmented methods supply extra context (Talaei et al., 2024; Shi et al., 2025) but trade recall for noise and suffer prompt dilution (Li et al., 2023). Multi-agent systems split roles (Wang et al., 2025; Askari et al., 2025; Cen et al., 2025) yet introduce coordination overhead and new failure modes (Pan et al., 2025). In both cases, arbitration often reverts to an LLM-as-a-judge (Gu et al., 2025), leaving the core reliability gap.

Seeking deterministic rather than probabilistic signals, recent work adopts debugger-style **post-execution validation** (e.g., SQL-PaLM (Sun et al., 2023b), SQL-CRAFT (Xia et al., 2024)), leveraging **database execution feedback (EF)** to guide LLM repairs. Although these engine-generated errors are faithful, they are inherently low-level: messages such as `Syntax error near WHERE` lack schema- and intent-level context, vary across dialects, and provide limited support for root-cause analysis. Moreover, EF offers no signal for semantically incorrect yet executable queries (e.g., wrong joins, or aggregation scopes) or for empty-result cases, leaving a substantial coverage gap. Conse-

quently, EF shifts rather than resolves the arbitration problem: **a probabilistic model still lifts coarse messages into high-level diagnoses and actionable repairs** (Chen et al., 2024). In the absence of a formal intermediate representation and a principled detection-and-repair procedure, such pipelines provide neither soundness (avoiding the introduction of new errors) nor completeness (exhaustively identifying statically-detectable violations), and lack a systematic bridge from diagnosis to repair. Hence, these observations expose a fundamental arbitration paradox: existing efforts delegate both diagnosis and repair decisions to the class of probabilistic models that produced the errors. This dependence on indeterminacy judgment creates a bottleneck that manifests as two challenges:

**Challenge 1: Diagnostic Translation Deficit.** Low-level diagnostics from execution feedback (EF) are engine-specific, local, and semantically opaque, offering limited support for root-cause analysis. What is missing is a deterministic procedure that lifts ambiguous, low-level observations into precise, schema- and intent-aware high-level diagnoses.

**Challenge 2: Repair Direction Deficit.** Even when precise semantic diagnoses are available, existing pipelines lack principled mechanisms to derive actionable, correctness-preserving repairs. Effective fixing must navigate the correctness–intent trade-off: preserving formal validity (syntax, types, and schema constraints) while maintaining the natural-language intent. This calls for a systematic method that compiles diagnostic information into constrained, verifiable repair instructions, providing soundness guarantees (no new violations), strong intent alignment, and, where applicable, completeness over statically detectable errors.

This paper introduces Post-SQLFix, a neuro-symbolic framework that shifts Text-to-SQL repair from ambiguous, feedback-driven heuristics to **formal, deterministic validation and synthesis**. We contend that the reliability of LLM-based SQL generation can be substantially improved by coupling neural semantic reasoning with a rigorous symbolic engine. To this end, we perform a two-stage query canonicalization that parses any SQL query into a dialect-aware, hierarchical **Canonical Query Structure (CQS)**, which serves as the intermediate representation and contract between the neural and symbolic components. Built on CQS, we implement a **hierarchical validator** that analyzes queries in a strictly ordered pipeline: Syntactic Analysis, Context-Free Semantic Analysis, and Context-Sensitive Semantic Analysis. This procedure yields precise, formal diagnoses of all statically detectable violations, providing soundness and, where applicable, completeness. Beyond diagnosis, each violation is compiled into a constrained synthesis problem. The symbolic engine enumerates a schema-grounded, finite set of repair plans that are provably valid with respect to the constraints and introduce no new violations. This design **repositions the LLM from an unreliable arbiter to a constrained neural agent** that realizes a formally vetted repair specification. The specification includes the diagnosis and the admissible repair plans, allowing the LLM to leverage its generative strengths while preserving formal guarantees for the final SQL. Our contributions are:

- We introduce a formal, neuro-symbolic framework for Text-to-SQL for repair that provides a sound and complete diagnostic engine, designed to supersede the limitations of Execution Feedback (EF).

- We formalize the repair process as a constrained synthesis problem, transforming EF's speculative, multi-turn repair loop into a single-pass, guided generation task.

- We demonstrate the empirical superiority and generalizability of our formal approach. On the BIRD and Spider benchmarks, including multi-dialect subsets, Post-SQLFix significantly improves execution accuracy by up to +11.6% and reduces repair iterations by 50% compared to the standard EF paradigm.

## 2 RELATED WORK

**Neuro-Symbolic Code Generation:** Recent neuro-symbolic approaches to code generation can be broadly categorized by their intervention point. Constrained decoding methods (Poesia et al., 2022; Mündler et al., 2025) enforce formal correctness by restricting the LLM's output space at the token level during generation. While achieving reduction in compilation errors (Mündler et al., 2025), this generation-time constraint paradigm remains limited to syntactic structure and local type constraints—it cannot address complex context-sensitive consistency errors where root causes are structurally distant from their manifestation points. SMT-solver-based verification approaches (Sun et al., 2023a) provide rigorous formal guarantees through satisfiability

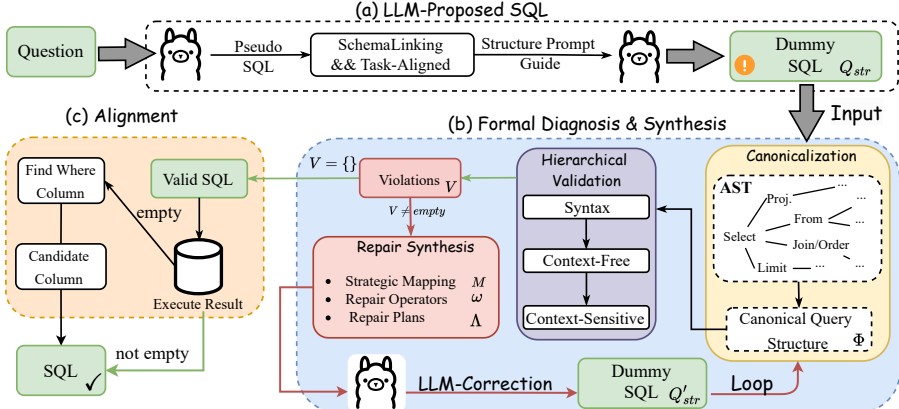

**Figure 1:** The framework consists of three main stages: (a) Initial LLM proposes a SQL query from user question. (b) Formal Diagnosis & Synthesis: the engine validates the query's formal correctness via three layers and synthesizes a valid repair plans for detection violations to guide LLM repair. (c) Alignment, a deterministic strategy for handling semantically ambiguous empty-result cases.

checking but face a critical *translation gap*: converting abstract satisfying assignments into semantically meaningful, intent-preserving code repairs still requires extensive manual prompt engineering. Similar challenges appear in neurosymbolic table extraction (Mehrotra et al., 2025), highlighting that this gap between formal correctness and practical repair represents a pervasive barrier. In contrast, our work performs holistic post-hoc diagnosis to systematically identify all error root causes, then automatically compiles this analysis into a formal, LLM-executable Repair Specification—inspired by GenSQL's symbolic representation framework (Huot et al., 2024). This diagnosis-driven synthesis paradigm enables provably sound handling of the complete error taxonomy while eliminating the translation gap.

**SQL Correction.** SQL repair methodologies can be broadly categorized. A significant work has focused on static analysis and rule-based methods to correct a limited set of common syntactic and logical errors (Presler-Marshall et al., 2021). While deterministic, these approaches often lack the flexibility to handle the diverse and complex errors generated by modern LLMs. In contrast, regardless of their architectural complexity, converge on a different fundamental dependency: **database execution feedback (EF)**. Post-execution validation methods (Sun et al., 2023b; Xia et al., 2024; Chen et al., 2023), multi-agent frameworks (Cen et al., 2025; Wang et al., 2025), and even self-correction paradigms (Pourreza & Rafiei, 2023; Askari et al., 2025) all universally rely on EF to trigger and guide the repair process. This reliance on EF exposes a critical limitation: database EF provide only coarse-grained *symptoms* rather than precise diagnostic information (Chen et al., 2024). While benchmarks like NL2SQL-BUGs (Liu et al., 2025) have established a rich taxonomy for these semantic errors, EF provides no direct path to identifying them. Consequently, these methods require LLMs to interpret ambiguous error signals, perpetuating rather than resolving the underlying probabilistic reasoning challenges, which often degrades performance (Huang et al., 2024; Kamoi et al., 2024). Our work moves beyond both template-application and feedback-driven speculation by introducing a formal synthesis paradigm grounded in a complete, semantic diagnosis.

## 3 METHODOLOGY

This section details the foundations of our work, designed to address the limitations of the prevailing database EF. The challenge lies in translating formal validation into actionable guidance for LLMs. Our objective is to bridge this neuro-symbolic divide by replacing EF's speculative repair loop with a constrained synthesis problem. To this end, as shown in Figure 1, we introduces a post-hoc repair pipeline rather than reinventing SQL generation (part a). This pipeline operates on two distinct tracks: a formal synthesis loop for incorrect queries (part b), complemented by a deterministic alignment strategy for ambiguous empty-result cases (part c). The entire framework is unified by its central abstraction, the Canonical Query Structure (CQS). The following sections will detail the CQS's formal definition (§3.1), its role in our hierarchical validation engine (§3.2), its function within our sound repair synthesis (§3.3-§3.4), finally, its application in the LLM-guidance workflow (§3.5).

| Validation Layer | Error Categories |
|---|---|
| **Syntactic** ($\alpha$) | This layer handles precursors to semantic analysis, such as typos (e.g., 'SEELCT') |
| **Context-Free Semantic** ($\beta$) | **Clause-related:** Clause Redundancy
**Function-related:** Aggregate Functions, Window Functions
**Operator-related:** Logical Operator (precedence issues)
**Other:** ASC/DESC, DISTINCT (structural usage) |
| **Context-Sensitive Semantic** ($\gamma$) | **Attribute-related:** Attribute Mismatch/Missing/Redundancy
**Table-related:** Table Mismatch/Missing/Redundancy
**Join-related:** Join Condition Mismatch, Join Type Mismatch
**Function-related:** Date/Time, Conversion, Math, String, Conditional (type/schema dependent usage)
**Value-related:** Data Format Mismatch |
| **Empty Result Warnings** | **Condition-related:** Explicit Condition Mismatch/Missing
**Value-related:** Value Mismatch
**Subquery-related:** Subquery Mismatch |

**Table 1:** Hierarchical Taxonomy of SQL Validation Errors

## 3.1 QUERY CANONICALIZATION

To overcome syntactic variance across SQL dialects (e.g., SQLite, PostgreSQL), our framework first parses any query string $Q_{str}$ into a dialect-aware **Abstract Syntax Tree (AST)**. While the AST unifies syntax, it is ill-suited for high-level semantic analysis regarding logical dependencies. We introduce a hierarchical representation, **Canonical Query Structure (CQS)**, designed to model a query's logical architecture. The CQS, denoted $\Phi$, is not merely a data structure but a formal abstraction that distills a query's essential operational components. We structure $\Phi$ as a 6-tuple:

$$\Phi = (\Pi, \rho, \Gamma, \Omega_{ord}, A, S). \tag{1}$$

This structure is explicitly designed to mirror the relational query evaluation process. It begins with the **Source Relation** $\rho$, which defines the intermediate working set. This relation is then transformed by the **Aggregation** logic $\Gamma$. The final result set is determined by the **Projection** $\Pi$ and formatted by the **Presentation** component $\Omega_{ord}$. This model is completed by components for managing **Aliases** ($A$) and recursively-defined **Nested Scopes** ($S$). The $S$ component decomposes complex structures (e.g., CTEs, derived tables) into independent logical units, providing a clean, structured basis for all subsequent formal analysis, independent of the query's original syntactic form. The detailed structural composition of each component is formally defined in Appendix C.1.

## 3.2 HIERARCHICAL VALIDATION ENGINE

As shown in Figure 1(b), the Post-SQLFix validation engine systematically inspects the CQS $\Phi$ against a hierarchy of integrity constraints. Our approach is grounded in the principles of formal language theory, mirroring a compiler's phased analysis. This process is structured into three validation layers with a partial ordering, denoted $\alpha \prec \beta \prec \gamma$ ensuring that foundational errors are resolved before dependent ones are considered. This strict partial ordering is foundational to our **diagnostic soundness**. By ensuring that foundational errors are resolved before dependent ones, this hierarchical dependency **guarantees zero false positives** arising from cascading failures.

**Syntactic Analysis (Layer $\alpha$).** This foundational layer validates the basic grammar of the query. This layer detects errors such as misspelled keywords or unbalanced parentheses.

**Context-Free Semantic Analysis (Layer $\beta$).** This layer verifies the **intrinsic logical integrity** of the CQS $\Phi$, independent of any external schema. It enforces universal constraints of relational logic that must hold true for any well-formed query, such as dependencies between clauses. In essence, this layer acts as an internal logic check, for example, it detects inconsistencies such as referencing an attribute in an `ORDER BY` clause that is not available in the `SELECT` list of the query.

**Context-Sensitive Semantic Analysis (Layer $\gamma$).** This layer validates the CQS $\Phi$ is semantically coherent with respect to the target database environment, $(\Sigma, \Delta)$, where $\Sigma$ is the database schema (defined in Appendix A.1) and $\Delta$ is the **Dialect Specification**. Analogous to a compiler's semantic analysis, this layer ensures **Identifier Binding** and **Type Correctness** against $\Sigma$. It also performs a **Structural Integrity Analysis** by verifying properties like join path connectivity using $\Sigma$'s foreign key constraints. Finally, it enforces dialect conformance, ensuring adherence to the function

signatures and rules defined in $\Delta$. Crucially, the validation process is recursive. While the trigger for checking nested scopes lies in Layer $\gamma$ (which manages context and visibility), the execution involves restarting the full semantic validation pipeline ($P_{\text{struct}} \wedge P_{\text{context}}$) for each nested unit. This treats every subquery or CTE as an independent CQS, ensuring that it satisfies all structural and schema constraints internally before being integrated into the outer query. The detailed formalization of these checks is provided in Appendix A.3.

This layered architecture provides a systematic engine for error detection. All examples of each layers are in Appendix F. We demonstrate that our hierarchy systematically covers the full spectrum of error categories cataloged by the NL2SQL-BUGs benchmark (Liu et al., 2025), as detailed in Table 1. Furthermore, our formalism extends beyond purely semantic taxonomies by incorporating the foundational syntactic layer ($\alpha$) and establishes an extensible framework where new validation rules can be integrated into the appropriate layer to support new SQL dialects or database functions.

### 3.2.1 Formal Correctness Specification

The hierarchical validation process culminates in a formal specification of a query's correctness. This specification operates on a **syntactically well-formed CQS** $\Phi$. The syntactic validation (Layer $\alpha$) acts as a gateway: if a query string $Q_{\text{str}}$ cannot be parsed into a CQS, it is rejected outright. For a successfully parsed query, we define its semantic correctness through two predicates corresponding to other validation layers:

- $P_{\text{struct}}(\Phi)$: The *Structural Correctness Predicate*, which holds true if the CQS $\Phi$ **and all its nested scopes** are internally consistent (Layer $\beta$).
- $P_{\text{context}}(\Phi, \Sigma, \Delta)$: The *Contextual Correctness Predicate*, which holds true if $\Phi$ recursively satisfies the database context constraints defined by the schema and the dialect (Layer $\gamma$).

A CQS $\Phi$ is deemed **semantically valid**, if and only if both predicates are satisfied:

$$\text{IS\_VALID}(\Phi, \Sigma, \Delta) \iff P_{\text{struct}}(\Phi) \wedge P_{\text{context}}(\Phi, \Sigma, \Delta) \tag{2}$$

A **Violation**, $v$, represents a specific instance of a semantic predicate failure. The set of all violations for a syntactically correct query is thus defined as:

$$V(\Phi, \Sigma, \Delta) = \{v \mid v \text{ is an atomic predicate that fails for } (\Phi, \Sigma, \Delta)\} \tag{3}$$

Consequently, a query is semantically invalid if and only if its violation set is non-empty.

### 3.3 A Formalism for Violation-Driven Repair

Given a CQS $\Phi$ with a non-empty violation set $V(\Phi, \Sigma, \Delta)$, the repair task is to synthesize a repair plan $\Lambda$ that transforms $\Phi$ into a valid $\Phi'$ for which $V(\Phi', \Sigma, \Delta) = \emptyset$. Our formalism models this as a constrained synthesis problem, built upon the concepts of repair operators and the plans they form. The entire process is designed to be violation-driven, ensuring that every synthesized action is a direct and relevant response to a detected invalidity.

### 3.3.1 The Building Blocks: Operators and Plans

Our repair formalism is built upon two core concepts: primitive **operators** that enact atomic changes, and **plans** that orchestrate sequences of these operators.

**Definition 1** (Repair Operator). *A Repair Operator, $\omega$, is a parametric transformation function $\omega : \Phi \to \Phi$ that applies a targeted, atomic modification to a Canonical Query Structure. The parameters of $\omega$ instantiate the operator for a specific application context.*

Since a single violation may require a sequence of modifications for a complete resolution, we introduce the concept of a Repair Plan, which defined as follow:

**Definition 2** (Repair Plan). *A Repair Plan, $\Lambda$, is an ordered sequence of instantiated repair operators, $\Lambda = \langle \omega_1, \omega_2, \ldots, \omega_n \rangle$. The application of a plan to a CQS, denoted $\Lambda(\Phi)$, is the functional composition of its constituent operators, $(\omega_n \circ \cdots \circ \omega_2 \circ \omega_1)(\Phi)$.*

These two constructs form the foundational vocabulary for our synthesis process, which we detail in the following section.

### 3.3.2 Violation-Driven Plan Synthesis

The challenge lies in synthesizing valid and semantically meaningful repair plans. Our framework avoids unconstrained search by employing a fully deterministic, symbolic violation-driven synthesis process. This entire process, detailed in Algorithm 1, is orchestrated by our symbolic engine and precedes any interaction with the LLM.

The process begins with a configurable strategic mapping, $M$, designed as an extensible interface. It associates each violation type with a set of candidate repair operator schemata known to address it:

$$M : v \rightarrow 2^{\omega_1, \omega_2, \cdots}. \tag{4}$$

For instance, for an `AttributeNotFound` violation, $M$ would map it to a set of operator schemata centered around $\omega_{replace}$. We provide a detailed instantiation of this mapping for key violation types in Table 14, and a Algorithm 1 first identifies the set of violations $V$. For each violation, it consults the mapping $M$ to retrieve and instantiate relevant operator schemata, which are then composed into candidate repair plans Finally, in a critical **verify-before-propose** step, each candidate plan is applied to $\Phi$, and the resulting CQS is checked against our correctness specification. Only those plans produce a verifiably valid successor state are included in the final output. By construction, this space contains a finite, diverse set of formally sound repair strategies, each guaranteed to resolve at least one initial violation. This entire process is designed for **repair completeness**: the strategic mapping $M$ is constructed to be a total function over the set of all detectable violations, ensuring a repair plan can always be synthesized for any error our identifies.

### 3.4 Soundness of Repair Synthesis

A critical property of our violation-driven synthesis process is its soundness, which guarantees that every generated repair plan, when applied, results in a formally correct query. This property is a direct consequence of the "verify-before-propose" step in our synthesis algorithm. We formalize this guarantee as a theorem. The procedure detailed in Algorithm 1 defines our *repair plan synthesis function*, denoted SYNTHESIZE_PLANS($\Phi, \Sigma, \Delta$). This function takes an invalid CQS $\Phi$ and returns a set of valid repair plans, $\{\Lambda\}$.

**Theorem 1** (Soundness of Repair Synthesis). *For any CQS $\Phi$ and its corresponding context $(\Sigma, \Delta)$, any repair plan $\Lambda \in$ SYNTHESIZE_PLANS($\Phi, \Sigma, \Delta$) will produce a semantically valid CQS when applied. That is:*

$$\forall \Lambda \in \text{SYNTHESIZE\_PLANS}(\Phi, \Sigma, \Delta), \quad \text{IS\_VALID}(\text{APPLY\_SEQUENCE}(\Phi, \Lambda), \Sigma, \Delta) \tag{5}$$

*Proof.* The proof follows directly from the construction of our synthesis algorithm, SYNTHESIZE_PLANS (§ 3.3.2). Specifically, the final verification step of the algorithm (Line 9–11 in Algorithm 1) explicitly checks each candidate plan $\Lambda_i$ by applying it and validating the resulting CQS with IS_VALID. The function is defined to only include those plans for which this check succeeds. A more detailed proof is provided in Appendix B. □

It provides a formal guarantee that **the LLM is only ever presented** with repair strategies that are provably correct. This guarantee eliminates the speculation inherent in feedback-driven methods, underpinning the reliability of our framework. While Theorem 1 establishes the soundness of single repair step, we extend this formal analysis to the entire iterative process in Appendix G and we prove **Order-Independent Reachability** (Theorem 8), demonstrating that finding *any valid repair sequence guarantees convergence to a correct state*.

### 3.5 Formal-Guided LLM Repair

The final stage of our framework leverages the formally synthesized repair space to guide a constrained LLM agent. This entire process is orchestrated by our symbolic engine, which we formalize as a top-level function.

**Definition 3** (Specification Creation). *Let*CREATE_SPECIFICATION *be the top-level function representing our symbolic engine. It takes a query string, $Q_{str}$, and produces either a formal **Repair Specification**, $S_R$, or a special symbol, `valid`. The function has the following signature:*

$$\text{CREATE\_SPECIFICATION} : Q_{str} \rightarrow S_R \cup \{valid\}$$

*Internally, this function orchestrates the full symbolic pipeline: it first performs **canonicalization** ($Q_{str} \rightarrow \Phi$) and **validation** ($\Phi \rightarrow V$). If violations are found, it proceeds to **plan synthesis** by invoking $\textsc{Synthesize\_Plans}(\Phi, \Sigma, \Delta)$ (from Algorithm 1) to generate a set of sound repair plans, $\{\Lambda\}$. These outputs are then composed into the **Repair Specification** ($S_R$), a tuple $\langle Q_{str}, V_{abs}, \Lambda_{abs} \rangle$ that provides the complete problem context for the LLM.*

With this definition, the neuro-symbolic repair process can be succinctly captured by the formulation:

$$Q'_{\text{str}} = \text{LLM}_{\text{agent}} \left( \textsc{Create\_Specification}(Q_{\text{str}}) \right) \tag{6}$$

The LLM is then tasked with a **constrained generation** objective: to produce a new, syntactically correct SQL query $Q'_{\text{str}}$ that accurately implements a transformation described by one of the plans in $\Lambda_{\text{abs}}$. This formulation explicitly models the neuro-symbolic synergy: the symbolic engine provides the formal **what** (the diagnosis) and **how** (the valid repair plans), while the neural agent provides the semantic **which** (the most plausible plan). This architecture effectively constructs a **safety guard** around the LLM. It is by *decoupling validity from semantic selection,* we ensure that the **output is bound by the reliability** guarantees of the symbolic system.

### 3.6 Alignment for Empty Results

A challenging scenario arises when a query is valid but returns an empty result. While an empty result can be the correct answer, it can also indicate a subtle semantic error where the query's logic does not align with the user's likely intent, particularly in question-answering contexts where a non-empty result is expected. For such cases, as shown in Figure 1(c), we employ a highly conservative repair strategy that prioritizes provable safety over speculative correction. We hypothesize a common cause is **Columns Mismatch** within a predicate in the query's filter conditions $F$. To preserve the semantic anchor provided by the literal value in the predicate, we seeks to identify an alternative attribute that would yield a non-empty result. We define a **Alignment Operator**, $\omega_{\text{align}}$.

**Definition 4** (Alignment Operator). *Given a CQS $\Phi$ and a target predicate $p = (c, \circ, v) \in F$, the operator $\omega_{align}(p)$ identifies a candidate attribute $c^*$ that satisfies the **data-evidence constraint**. This constraint requires that there must exist at least one tuple in the database instance $D$ for which the original predicate logic holds true with the new attribute: $\exists\ tuple \in D$ such that $tuple[c^*] \circ v$*

The application of this operator is strictly constrained. First, the search for $c^*$ is limited to attributes within the relations $T_\rho$ of the CQS $\Phi$. Second, the operator is only applied if a viable $c^*$ is **uniquely** identified, preventing ambiguous repairs. If these conditions are met, the engine synthesizes and executes a single repair plan $\Lambda = \langle \omega_{\text{replace}}(c, c^*) \rangle$. Otherwise, no action is taken. This design guarantees that the alignment process **never introduces any errors**, a claim we empirically validate in § 4 and Table 7. The details of the algorithm are in Appendix C.5.

| Method | Inference Model | Zero-shot Setting | Retrieval | Accuracy (%) | | | |
|---|---|---|---|---|---|---|---|
| | | | | Simple | Moderate | Challenging | All |
| Gen-SQL (Shi et al., 2025) | Qwen2.5-Coder-7B | ✗ | ✓ | 53.4 | 34.9 | 26.9 | 45.3 |
| Gen-SQL (Shi et al., 2025) | Qwen2.5-Coder-32B | ✗ | ✓ | 61.6 | 42.2 | 35.9 | 53.3 |
| TA-SQL (Qu et al., 2024) | Qwen2.5-Coder-7B | ✓ | ✗ | 54.70 | 33.41 | 29.66 | 45.89 |
| TA-SQL (Qu et al., 2024) | Qwen2.5-Coder-32B | ✓ | ✗ | 66.70 | 49.14 | 46.21 | 59.45 |
| Gen-SQL + Post-SQLFix | Qwen2.5-Coder-7B | ✗ | ✓ | **63.5**↑$_{+10.1}$ | **44.0**↑$_{+9.1}$ | **35.2**↑$_{+8.3}$ | **54.9**↑$_{+9.6}$ |
| Gen-SQL + Post-SQLFix | Qwen2.5-Coder-32B | ✗ | ✓ | **70.8**↑$_{+9.2}$ | **57.1**↑$_{+14.9}$ | **51.7**↑$_{+15.8}$ | **64.9**↑$_{+11.6}$ |
| TA-SQL + Post-SQLFix | Qwen2.5-Coder-7B | ✓ | ✗ | **64.00**↑$_{+9.3}$ | **46.98**↑$_{+13.57}$ | **38.62**↑$_{+8.96}$ | **56.45**↑$_{+10.56}$ |
| TA-SQL + Post-SQLFix | Qwen2.5-Coder-32B | ✓ | ✗ | **70.49**↑$_{+3.8}$ | **56.03**↑$_{+6.9}$ | **50.34**↑$_{+4.1}$ | **64.21**↑$_{+4.8}$ |

**Table 2:** Execution Accuracy on BIRD Development Dataset.

## 4 Experiment

### 4.1 Experimental Setup

All experiments are conducted on Ubuntu 22.04 equipped with 64GB RAM and 32-core Intel 5.0GHz CPU . Open-source LLMs are deployed locally using 8*80GB GPUs with BF16 precision. All reported results are averaged 5 runs with a standard deviation of 0.9% in Gen-SQL and 0.7% in TA-SQL to ensure statistical significance. All prompt details are provided in Appendix H

| Method | Inference Model | Zero-shot Setting | Retrieval | Accuracy (%) | | | |
|---|---|---|---|---|---|---|---|
| | | | | Simple | Moderate | Challenging | All |
| Gen-SQL (Shi et al., 2025) | Qwen2.5-Coder-7B | × | ✓ | 82.6 | 57.5 | 50.0 | 72.1 |
| Gen-SQL (Shi et al., 2025) | Qwen2.5-Coder-32B | × | ✓ | 82.7 | 58.0 | 50.6 | 72.4 |
| TA-SQL (Qu et al., 2024) | Qwen2.5-Coder-7B | ✓ | × | 78.0 | 59.8 | 34.7 | 65.1 |
| TA-SQL (Qu et al., 2024) | Qwen2.5-Coder-32B | ✓ | × | 81.3 | 71.8 | 47.1 | 72.1 |
| Gen-SQL + Post-SQLFix | Qwen2.5-Coder-7B | × | ✓ | $83.8\uparrow_{+1.2}$ | $60.9\uparrow_{+3.4}$ | $52.4\uparrow_{+2.4}$ | $73.9\uparrow_{+1.8}$ |
| Gen-SQL + Post-SQLFix | Qwen2.5-Coder-32B | × | ✓ | $87.2\uparrow_{+4.5}$ | $67.2\uparrow_{+9.2}$ | $57.1\uparrow_{+6.5}$ | $77.6\uparrow_{+5.2}$ |
| TA-SQL + Post-SQLFix | Qwen2.5-Coder-7B | ✓ | × | $79.5\uparrow_{+1.5}$ | $65.5\uparrow_{+5.7}$ | $47.1\uparrow_{+12.4}$ | $69.5\uparrow_{+4.4}$ |
| TA-SQL + Post-SQLFix | Qwen2.5-Coder-32B | ✓ | × | $83.2\uparrow_{+1.9}$ | $72.4\uparrow_{+0.6}$ | $54.7\uparrow_{+7.6}$ | $75.0\uparrow_{+2.9}$ |

**Table 3:** Execution Accuracy on Spider Development Dataset.

| Metric | Strategy | Generation Method | |
|---|---|---|---|
| | | Gen-SQL | TA-SQL |
| Accuracy (%) | N/A | 45.31 | 45.89 |
| | Reflection | 47.07 | 46.54 |
| | EF | 49.88 | 51.00 |
| | **Our** | **53.13** | **53.32** |
| Left Errors | N/A | 340 | 306 |
| | Reflection | 291 | 280 |
| | EF | 72 | 60 |
| | **Our** | **35** | **32** |

**Table 4:** Performance Comparison of SOTA Methods on BIRD (Li et al., 2024a) using Qwen2.5-Coder-7B.

| Metric | Strategy | Generation Method | |
|---|---|---|---|
| | | Gen-SQL | TA-SQL |
| Accuracy (%) | EF | 49.88 | 51.00 |
| | **Our** | **53.13** | **53.32** |
| Tokens (avg) | EF | 1,867 | 2,409 |
| | **Our** | **1,506** | **1,679** |
| Time (s) | EF | 19.53 | 17.68 |
| | **Our** | **14.56** | **14.91** |
| Rounds (avg) | EF | 2.00 | 2.81 |
| | **Our** | **1.30** | **1.40** |

**Table 5:** Efficiency-Accuracy Analysis for SQL Repair Methods in BIRD (Li et al., 2024a) Dataset

**Datasets and Metrics** We evaluate our framework on three benchmarks: BIRD (Li et al., 2024a), chosen for its real-world complexity, Spider (Yu et al., 2018), for its large-scale, cross-domain diversity datasets, and BIRD Mini-Dev(Li et al., 2024b), chosen for its multi-dialects (PostgreSQL and MySQL). Our evaluation encompasses 5 key metrics: execution accuracy, token efficiency, time efficiency, repair rounds, and residual error count. These metrics collectively assess both effectiveness and computational efficiency of our approach and evaluate with the official tools.

**Implementation** Our framework is written in over 8k lines of Python code We employ the code generation models: Qwen2.5-Coder-7B and 32B as our LLM backbone. The hierarchical rules and repair specifications are implemented according to the framework, with complete details provided in Appendix C. Temperature is set to 0.8 for balanced exploration. It is crucial to note: our selection is not driven by pursuing SOTA execution accuracy on leader boards. Instead, we chose Gen-SQL and TA-SQL framework as front-end, yet accessible open-source and reproduction models and architectures to demonstrate the *universal efficacy and plug-and-play* of Post-SQLFix. Our contribution lies in the **relative improvement** that Post-SQLFix brings to a given model, rather than the absolute final score. Three case studies are shown in Appendix H.

### 4.2 EXPERIMENTAL RESULTS

#### 4.2.1 RQ1: FUNDAMENTAL EFFICACY ANALYSIS:

Our framework's hierarchical design enables two distinct modes of operation: as a deeply integrated validation layer within a generation pipeline, and as a post-hoc repair module.

**Universal Efficacy and Integration:** A key strength of Post-SQLFix is architectural universality. The formalizability of database schemas enables our system to capture and rectify any database-related errors in LLM outputs at the earliest possible stage, before propagation to downstream components. As demonstrated in Tables 2 and 3, our framework delivers substantial and consistent accuracy improvements when integrated into a generation pipeline. This holds true across a wide range of setups, including diverse base models and various inference settings such as zero or few-shot, with and without retrieval, achieving gains as high as +11.6% on BIRD.

**Plug-and-Play:** Beyond this integrated performance, we also verified its utility as a post-hoc repair module (Table 4). In this capacity, it still boosts baseline accuracies by over 7% without requiring any architectural modifications.

| Configuration | Accuracy (%) | Errors Left | Empty Results | Tokens (avg) |
|---|---|---|---|---|
| Baseline (TA-SQL-7B Generated) | 45.89 | 306 | 85 | – |
| *7B* Post-SQLFix (Full System) | **53.32** | **11** | 100 | **2,034** |
| *7B* - w/o Hierarchical Validation | 52.26 | 39 | 100 | 2,426 |
| *7B* - w/o Plan Synthesis | 50.24 | 52 | 101 | 2,916 |
| *32B* Post-SQLFix (Full System) | **54.11** | **9** | 91 | **1,113** |
| *32B* - w/o Hierarchical Validation | 52.96 | 15 | 95 | 1,306 |
| *32B* - w/o Plan Synthesis | 51.26 | 28 | 93 | 1,502 |

**Table 9:** Ablation Study: Component Contributions to Post-SQLFix Performance.

The performance gap between the universal and post-hoc approaches (e.g., 54.9% vs. 53.1% for Gen-SQL) reveals an insight: *integrated hierarchical validation is fundamentally superior to terminal correction*. By intercepting and resolving errors early, our framework prevents their downstream propagation, yielding compounding benefits that a simple post-hoc repair tool cannot achieve.

**Robustness to Instruction Non-Compliance.** We conducted a targeted analysis on the TA-SQL repair process for the BIRD dataset to quantify instances of LLM non-compliance. A "non-compliant error" is defined as a case where the LLM, despite being given a correct and complete repair specification ($S_B$), generates a query that either fails to implement the plan or introduces new formal errors. We tested several code generation models, with results summarized in Table 6. *Non-compliance* refers to the number of LLMs does not follow the instructions or lost any semantics during repairing process; *Capture Rate* is the percentage of these errors caught by the validation loop (Figure 1b).

The results reveal two key findings. First, non-compliance is rare but does occur, particularly with smaller or less mature models, while the powerful Qwen2.5-32B model exhibited perfect instruction-following. Second, and most importantly, for every single instance of non-compliance, our framework's post-generation validation loop successfully captured the error, preventing an incorrect query from being finalized.

| Metric | Qwen2.5 -7B | Qwen2.5 -32B | Codestral -7B | gpt-oss -120B | Llama-3.1 -70B | Mistral -24B |
|---|---|---|---|---|---|---|
| EF Accuracy (%) | 51.00 | 52.00 | 51.37 | 48.04 | 48.37 | 48.24 |
| Our Accuracy (%) | 53.32 | 54.11 | 52.80 | 53.19 | 53.89 | 54.37 |
| Non-compliance | 2 | 0 | 2 | 15 | 15 | 1 |
| Capture Rate | 100% | NA | 100% | 100% | 100% | 100% |

**Table 6:** Analysis of LLM Instruction Non-Compliance on BIRD repairs.

### 4.2.2 RQ2: COMPARATIVE ANALYSIS

We conduct evaluation of Post-SQLFix vs. Execution-Feedback (EF), the established post-validation repair standard, across three dimensions: accuracy, efficiency, and error resolution capability.

**Accuracy and Efficiency:** Table 4 demonstrates Post-SQLFix's consistent accuracy improvements over EF: +4.2% for Gen-SQL and +3.9% for TA-SQL. To ensure fair comparison, Table 5 evaluates both methods on identical repairable query sets. We achieve substantial efficiency gains: 19.3% token reduction for Gen-SQL, 30.3% for TA-SQL, and up to 50% fewer debugging iterations. By resolving all syntactic errors in a single, deterministic, LLM-free pass, we completely eliminate what would otherwise be a potentially lengthy and unreliable iterative loop with an LLM. This allows the system to immediately focus on the more challenging semantic errors, leading to faster convergence on a correct query.

| Metric | TA-SQL | EF | Alignment-LLM | Ours |
|---|---|---|---|---|
| Accuracy (%) | 45.89 | 46.09 | 45.96 | 46.09 |
| Empty Results | 85 | 73 | 57 | 82 |
| Time (s) | – | 1500 | 1046 | **45** |
| Syntax Errors | 306 | 312 | 327 | **306** |

**Table 7:** Performance Comparison of Different Alignment in BIRD. Alignment-LLM refers the symbolic engine generate candidate repairs and then the LLM judge the intent.

**Error Resolution Analysis.** Table 8 reveals Post-SQLFix's superior semantic error handling. While both methods achieve perfect syntactic error elimination, Post-SQLFix shows 50% better context-free error resolution and 56% improvement in context-sensitive errors. More residual analysis is in Appendix D.2.

| Error Type | Reflection | EF | Our-7B | Our-32B |
|---|---|---|---|---|
| Syntactic | 61 | 0 | **0** | **0** |
| Context-Free | 17 | 8 | **4** | 2 |
| Context-Sensitive | 200 | 64 | **28** | 7 |

**Table 8:** Residual Error (generated by TA-SQL in BIRD dataset) Distribution

**Empty Result Handling.** To analyze our strategy for ambiguous empty-result cases, we compare our method against both the standard EF paradigm and a flexible Alignment-LLM baseline, where an LLM selects from our formally-vetted candidate columns. Table 7 reveals a critical trade-off. Alignment-LLM is most aggressive in reducing 33% empty results (from 85 to 57), but this comes at the cost of reliability, as it **introduces 21 new query errors**. In contrast, our approach matches the accuracy of the much slower EF baseline while being **over 30 × faster**. Most importantly, unlike both EF and Alignment-LLM which introduce new errors, we can guarantees **a zero error introduction**. This result empirically validates our design choice: prioritizing the provable safety and efficiency essential for real-world systems over the speculative and unreliable gains from probabilistic repairs.

**Multi-Dialect Analysis.** The results, presented in Table 10, demonstrate our framework's superiority is its performance on dialects like MySQL and PostgreSQL. While richer feedback offers a marginal benefit to EF's accuracy, our approach not only achieves a higher final accuracy but does so with an **order-of-magnitude greater efficiency**. Notably, EF's token consumption escalates dramatically on these dialects (from 1.8k on SQLite to over 24k on MySQL), likely due to the LLM's unfamiliarity with their specific error highlights a critical vulnerability of feedback-driven methods. This finding confirms our central thesis: the limitation of EF is architectural, not merely informational. An informational limitation could be solved by providing more data, such as a more verbose error message. Our results show this yields only marginal gains. The core problem is architectural: the EF paradigm forces a probabilistic LLM to act as a diagnostician based on textual symptoms. Even the most detailed description remains a symptom that requires unreliable speculation. In contrast, our framework's architecture employs a symbolic engine that acts like a diagnostic tool. It doesn't guess based on the symptom; it performs a deterministic, root-cause analysis of the query's formal structure and provides a provably correct treatment plan.

| Method | Acc.(%) | Tokens (avg) | Time(s) | Errors |
|--------|---------|--------------|---------|--------|
| *MySQL* | | | | |
| Base | 41.0 | – | – | 68 |
| EF | 44.0 | 24070 | 7200 | 23 |
| **Ours** | **45.0** | **1553** | **888** | **15** |
| *PostgreSQL* | | | | |
| Base | 36.4 | – | – | 117 |
| EF | 38.0 | 41670 | 4323 | 62 |
| **Ours** | **41.0** | **10336** | **1378** | **30** |

**Table 10:** Performance Comparison with EF in Mini-dev for multi-dialect. The "Base" score represents the official benchmark result obtained by testing.

### 4.2.3 RQ3: Component Ablation Analysis:

Table 9 presents results on the same baseline queries (7B-generated) with controlled removal of key framework components. For Component Contribution Analysis, both validations and Formal-Guided Repair demonstrate substantial individual contributions. Removing Hierarchical Diagnosis results in accuracy drops of 1.00% (7B repair) and 1.15% (32B repair), with corresponding increases in residual errors (32→39 and 9→15 respectively). Removing Formal-Guided Repair causes more severe degradation: 3.02% accuracy loss (7B) and 2.85% (32B), with errors increasing to 52 and 28 respectively. This pattern indicates that while both components are essential, the repair strategy provides greater performance impact than validation refinement. Regarding the influence of LLMs on the framework, we also conducted experiments. The result of scaling effects are in Appendix D.1.

## 5 Conclusion

This paper addressed the reliability bottleneck in Text-to-SQL: the architectural limitations of the Execution-Feedback (EF) paradigm, which relies on probabilistic models to interpret ambiguous, symptomatic error signals. We introduced Post-SQLFix, a neuro-symbolic framework that instigates a paradigm shift from ambiguous feedback to **formal, deterministic synthesis**. By systematically separating verifiable diagnosis and repair synthesis (the symbolic engine) from semantic, intent-driven selection (the neural agent), our framework transforms the LLM from an unreliable arbiter into a constrained, formally-grounded agent. Our evaluation validates this new paradigm, demonstrating that Post-SQLFix delivers substantial improvements in accuracy and efficiency. By establishing a principled foundation for correctness and providing a blueprint for robust neuro-symbolic synergy, our work represents a significant step towards building trustworthy AI-driven code generation systems.

## 6 ETHICS STATEMENT

In accordance with the Code of Ethics, we have considered the broader impacts of our work. We already open all sources code. The primary positive societal contribution of this research is the enhancement of reliability and trustworthiness in AI-driven code generation systems. By providing a framework that can formally verify and correct SQL queries, our work can help mitigate the risks associated with deploying LLMs in data-critical applications, such as healthcare, finance, and scientific research, where an incorrect query could lead to flawed analysis or harmful decisions. This contributes to the responsible stewardship of AI by establishing a principled foundation for correctness.

## 7 REPRODUCIBILITY STATEMENT

To facilitate and promote future research, we have made our implementation and all associated data publicly available at Anonymous (2025). We will also submit our artifact for evaluation upon acceptance.

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

## A    DETAILED FORMALIZATION

### A.1    DATABASE SCHEMA REPRESENTATION

We represent a database schema as the 5-tuple:

$$\Sigma = (\mathcal{T}, \mathcal{C}, \mathcal{K}_{\text{pk}}, \mathcal{K}_{\text{fk}}, \Theta), \tag{7}$$

where $\mathcal{T}$ is a set of table names and $\mathcal{C}$ is the set of all fully qualified columns $c = (t, c)$ with $t \in \mathcal{T}$ and $c \in \mathcal{C}$. Let $table(c)$ be the projection of its table. The set of primary key columns is $\mathcal{K}_{\text{pk}} \subseteq \mathcal{C}$, and foreign key relationships are a set of pairs $\mathcal{K}_{\text{fk}} \subseteq \mathcal{C} \times \mathcal{C}$. Finally, $\Theta : \mathcal{C} \to$ Type maps each column to its data type.

### A.2    CANONICAL QUERY STRUCTURE (CQS)

**Definition 5** (Canonical Query Structure). *A SQL query Q is represented by a CQS, a hierarchical structure $\Phi$ that models the query's logical architecture. It is defined as a primary tuple:*

$$\Phi = (\Pi, \rho, \Gamma, \Omega_{ord}, A, S)$$

*where the components are logically grouped as follows:*

- *$\Pi$: The **Projection Specification**, a set $C$ of projected attributes (columns). Each attribute is fully qualified with its source relation or alias.*

- *$\rho$: The **Data Source Specification**, a tuple $(T, J, F)$ defining the intermediate working relation:*

  - *$T$: A set of base relations (tables).*
  - *$J$: A set of join predicates.*
  - *$F$: A predicate tree for the WHERE clause selection conditions.*

- *$\Gamma$: The **Aggregation Specification**, a tuple $(G, H)$ defining the grouping and aggregation logic:*

  - *$G$: An ordered set of grouping attributes.*
  - *$H$: A predicate tree for the HAVING clause conditions.*

- *$\Omega_{ord}$: The **Presentation Specification**, a tuple $(O, L)$ defining the final result formatting:*

  - *$O$: An ordered list of sorting specifications.*
  - *$L$: An integer for the cardinality limit (LIMIT).*

- *$A$: A set of **Alias Assignments**, a cross-cutting component mapping relations or attributes to their aliases for identifier resolution.*

- *$S$: A set of **Nested Scopes**. This component manages recursive logical units (e.g., CTEs, derived tables, subqueries). Each scope is recursively defined as an independent CQS $\Phi'$, augmented with its contextual role, enabling modular verification of arbitrary nesting depths.*

### A.3    FORMALIZATION OF LAYER $\gamma$

The Context-Sensitive Semantic Analysis layer ($\gamma$) verifies a structurally consistent CQS $\Phi$ against a target environment $(\Sigma, \Delta)$. Crucially, this process is defined recursively to handle nested scopes (CTEs, subqueries). Its corresponding predicate, $P_{\text{schema}}(\Phi, \Sigma, \Delta)$, is a conjunction of five sub-predicates:

$$P_{\text{schema}}(\Phi, \Sigma, \Delta) \Leftrightarrow P_{\text{bind}} \wedge P_{\text{type}} \wedge P_{\text{integrity}} \wedge P_{\text{dialect}} \wedge P_{\text{recurse}}$$

**Identifier Binding ($P_{\textbf{bind}}$)** This predicate ensures that all identifiers used within the current scope of CQS $\Phi$ are validly defined in the schema $\Sigma$ or local aliases.

$$P_{\text{bind}}(\Phi, \Sigma) \Leftrightarrow (\forall t \in T_\rho, t \in \Sigma.\mathcal{T}) \wedge (\forall c \in \text{AllAttributes}(\Phi), \exists t \in T_\rho \text{ s.t. } (t, c) \in \Sigma.\mathcal{C})$$

This check is fundamental for resolving all identifiers before further analysis.

**Type Correctness ($P_{\textbf{type}}$)** This predicate verifies that all operations are type-safe. For every operation within $\Phi$'s expressions, it checks:

$$P_{\text{type}}(\Phi, \Sigma) \Leftrightarrow \forall \text{op}(\mathbf{a}) \in \mathcal{O}(\Phi) : \text{COMPATIBLE}(\text{op}, \tau(\mathbf{a})) \tag{8}$$

where:

- $\mathcal{O}(\Phi)$ denotes the set of all operation expressions in the CQS $\Phi$.
- $\mathbf{a}$ represents the argument vector $\langle a_1, \ldots, a_n \rangle$ for the operation.
- $\tau(\mathbf{a})$ applies the type resolution function $\tau(x)$ to each argument in the vector.
- COMPATIBLE is a predicate that returns true if the operation signature matches the argument types.

**Structural Integrity Analysis ($P_{\textbf{integrity}}$)** This predicate enforces deeper semantic integrity rules derived from the schema structure . A primary example is **Join Path Validation**: We construct a connectivity graph $G_\Sigma = (T_{\text{db}}, K_{\text{fk}})$ from the schema's tables and foreign key relationships. This check verifies that the subgraph induced by the set of tables $T_P$ in the query is a single connected component in $G_\Sigma$. This rule is a powerful heuristic for preventing heuristically prevents semantically unintended cartesian products.

**Dialect Conformance ($P_{\textbf{dialect}}$)** This predicate enforces rules specific to the target SQL dialect, $\Delta$. Crucially, its validation depends only on the query $\Phi$ and the dialect specification $\Delta$, not the specific user schema $\Sigma$.

**Recursive Scope Validation ($P_{\textbf{recurse}}$)** To enable the "Recursive Decomposition" capability claimed in our framework, this predicate enforces that all nested logical units (e.g.,CTEs, derived tables) defined in component $S$ of $\Phi$ are themselves valid.

$$P_{\text{recurse}}(\Phi, \Sigma, \Delta) \Leftrightarrow \forall \Phi' \in \Phi.S : \text{IS\_VALID}(\Phi', \Sigma, \Delta) \tag{9}$$

This recursive definition ensures that validation is not merely top-level but penetrates every logical unit. Note that by invoking IS\_VALID, this predicate implicitly enforces a full re-evaluation of Layer $\beta$ and Layer $\gamma$ rules for the nested scope, effectively creating a validation loop that resets for each structural depth.

## B  FORMAL GUARANTEES OF THE POST-SQLFIX FRAMEWORK

This appendix provides a formal analysis of the soundness and completeness properties of our hierarchical validation and repair framework. We establish theoretical guarantees that our system produces no false positives in error detection (soundness of diagnosis) and ensures all synthesized repairs are formally correct (soundness of repair). The analysis builds upon the foundational concepts established in Section 3.

### B.1  FOUNDATIONAL DEFINITIONS

We begin by formalizing the set of valid queries based on the correctness specification from Section 3.2.1.

**Definition 6** (The Set of Valid CQS). *Given a database schema $\Sigma$, the set of all valid Canonical Query Structures, denoted $\Phi_{valid}(\Sigma)$, is the set of all CQS $\Phi$ for which our correctness specification holds true:*

$$\Phi_{valid}(\Sigma) = \{\Phi \mid \text{IS\_VALID}(\Phi, \Sigma) = true\} \tag{10}$$

*A query string $Q_{str}$ is considered valid if and only if it can be successfully parsed into a CQS $\Phi$ such that $\Phi \in \Phi_{valid}(\Sigma)$.*

**Definition 7** (Violation Detection Function). *Our violation detection function, $V(\Phi, \Sigma)$, as defined in Equation 3, returns the set of all predicate failures for a given CQS $\Phi$.*

**Definition 8** (Repair Synthesis Function). *Our repair synthesis function, $\text{SYNTHESIZE\_PLANS}(\Phi, \Sigma)$, takes an invalid CQS $\Phi$ and a schema $\Sigma$ as input. It generates the Constrained Repair Space, which is a set of valid Repair Plans:*

$$\text{SYNTHESIZE\_PLANS}(\Phi, \Sigma) = \{\Lambda \mid \Lambda(\Phi) \in \Phi_{valid}(\Sigma)\} \tag{11}$$

*The function returns an empty set if no valid repair plan can be synthesized.*

### B.2 SOUNDNESS OF DIAGNOSIS

**Theorem 2** (Soundness of Violation Detection). *Our violation detection framework is sound. For any CQS $\Phi$ and schema $\Sigma$, if our engine detects violations, then the CQS is not valid:*

$$V(\Phi, \Sigma) \neq \emptyset \Rightarrow \Phi \notin \Phi_{valid}(\Sigma) \tag{12}$$

*Proof.* The proof follows directly from the definitions. By Definition 6, a CQS $\Phi$ is in $\Phi_{\text{valid}}(\Sigma)$ if and only if $\text{IS\_VALID}(\Phi, \Sigma)$ is true. By the definition of our violation set $V$ (Equation 3), $V(\Phi, \Sigma)$ is non-empty if and only if there is at least one predicate failure in $\neg\text{IS\_VALID}(\Phi, \Sigma)$.

Therefore, the condition $V(\Phi, \Sigma) \neq \emptyset$ is logically equivalent to $\neg\text{IS\_VALID}(\Phi, \Sigma)$, which by definition implies $\Phi \notin \Phi_{\text{valid}}(\Sigma)$. This guarantees that our framework produces no false positives; every reported violation corresponds to a genuine failure to meet the correctness specification. $\square$

### B.3 SOUNDNESS OF REPAIR

**Theorem 3** (Soundness of Repair Plan Synthesis). *Our repair synthesis framework is sound. For any CQS $\Phi$ and schema $\Sigma$, any repair plan $\Lambda$ generated by our synthesis function produces a valid CQS:*

$$\forall\Lambda \in \text{SYNTHESIZE\_PLANS}(\Phi, \Sigma), \Lambda(\Phi) \in \Phi_{valid}(\Sigma) \tag{13}$$

*Proof.* The proof follows from the construction of the synthesis algorithm described in § 3.3.2. The final step of the algorithm (Step 3: Plan Composition and Verification) explicitly verifies each candidate repair plan $\Lambda_i$ by checking if its application results in a valid CQS: $\text{IS\_VALID}(\Lambda_i(\Phi), \Sigma)$.

The synthesis function $\text{SYNTHESIZE\_PLANS}(\Phi, \Sigma)$ is defined to only include those plans $\Lambda_i$ that successfully pass this verification step. Therefore, by definition, any plan $\Lambda$ returned by the function is guaranteed to produce a CQS $\Phi' = \Lambda(\Phi)$ such that $\text{IS\_VALID}(\Phi', \Sigma)$ holds. This implies $\Phi' \in \Phi_{\text{valid}}(\Sigma)$.

This **verify-before-propose** principle ensures that the LLM is only ever presented with repair strategies that are formally guaranteed to be correct, thus ensuring the soundness of the entire repair process. $\square$

### B.4 COMPLETENESS CONSIDERATIONS

Our framework provides strong, formally-defined completeness guarantees with respect to its operational scope. We distinguish between the completeness of our diagnostic capabilities and our repair synthesis capabilities.

### B.4.1 Diagnostic Completeness

We claim that our validation engine is complete for the class of all statically-detectable errors, as formally defined by a comprehensive external taxonomy.

**Theorem 4** (Diagnostic Completeness). *Let $\mathcal{E}_{bugs}$ be the set of all error categories defined in the NL2SQL-BUGs taxonomy. For any query $Q_{str}$ that contains an error $e \in \mathcal{E}_{bugs}$, its corresponding CQS $\Phi$ (if parsable) will have a non-empty violation set:*

$$\forall e \in \mathcal{E}_{bugs}, HasError(Q_{str}, e) \implies V(\Phi, \Sigma) \neq \emptyset$$

*Proof Sketch.* The proof proceeds by construction, demonstrating a complete mapping from the error categories in $\mathcal{E}_{\text{bugs}}$ to the validation rules within our three-layer hierarchy.

- **Syntactic Errors** (e.g., misspelled keywords, syntax errors from NL2SQL-BUGs) are guaranteed to be detected by **Layer** $\alpha$, as they will cause the AST parsing to fail, preventing the formation of a valid CQS $\Phi$.

- **Context-Free Semantic Errors** (e.g., `GROUP BY` clause inconsistencies, invalid use of operators) are systematically detected by the universal relational logic rules in **Layer** $\beta$. The constraints enforced by $P_{\text{struct}}(\Phi)$ are explicitly designed to cover these internal consistency issues.

- **Context-Sensitive Semantic Errors** (e.g., incorrect table/column names, invalid join paths, type mismatches) are comprehensively covered by the validation rules in **Layer** $\gamma$. The $P_{\text{schema}}(\Phi, \Sigma)$ predicate performs exhaustive checks of all identifiers and operations against the ground-truth schema $\Sigma$.

A detailed table mapping each specific NL2SQL-BUGs category to its corresponding validation rule in our framework is provided in Table 1. This demonstrates that no statically-detectable error defined by this comprehensive benchmark can pass our validation engine undetected. $\square$

### B.4.2 Repair Completeness

We claim that for any error our engine can diagnose, it can also propose a formally sound solution.

**Proposition 1** (Repair Completeness). *The repair synthesis process is complete with respect to the set of detectable violations. For any CQS $\Phi$ and schema $\Sigma$ such that $V(\Phi, \Sigma) \neq \emptyset$, the synthesized set of repair plans is non-empty:*

$$V(\Phi, \Sigma, \Lambda) \neq \emptyset \implies \text{Synthesize\_Plans}(\Phi, \Sigma, \Lambda) \neq \emptyset$$

*Argument by Design.* This property is a direct consequence of the design of the strategic mapping $M$ (Equation 4). $M$ is constructed to be a total function over the domain of all detectable violation types $v$. For every violation $v$ that can be identified by our validation predicates $P_{\text{struct}}$ and $P_{\text{schema}}$, a corresponding entry in $M$ defines at least one valid operator schema $\omega$. As the synthesis process instantiates these schemata, it is guaranteed to generate at least one candidate repair plan. Therefore, the completeness of repair is an architectural guarantee of the framework. $\square$

### B.4.3 Formal Limitation: Intent-Level Completeness

While our framework guarantees formal correctness, we do not claim **intent-level semantic completeness**. Ensuring that a formally valid query perfectly matches a user's unstated intent is formally undecidable and remains beyond the scope of any static analysis framework. Our goal is to guarantee formal correctness, which is a necessary precondition for achieving intent alignment.

## C  Detailed Formalization of the Validation Engine

This appendix provides a detailed, operational formalization of the hierarchical validation engine described in § 3.3.1. We specify the structural composition of the Canonical Query Structure (CQS), define necessary auxiliary functions, and present the concrete validation rules.

## C.1 STRUCTURAL COMPOSITION OF THE CQS

The Canonical Query Structure $\Phi$ introduced in Definition 5 is composed of several components, whose detailed structures are defined as follows. The **Source Relation** $\rho$ is a tuple $(T, J, F)$:

- $T$: A set of source table tuples (table_name, alias), where alias can be NULL.
- $J$: A set of join tuples (join_type, table_spec, predicate), where join_type includes INNER, LEFT, etc.
- $F$: A set of predicates from the WHERE clause.

The **Projection** $\Pi$ is a set of tuples (expression, alias) from the SELECT clause. The **Aggregation** $\Gamma$ is $(G, H)$, where $G$ is a set of grouping expressions and $H$ is a set of HAVING predicates. The **Presentation** $\Omega_{\text{ord}}$ is $(O, L)$, where $O$ is a set of ordering expressions and $L$ is the limit.

This detailed representation explicitly captures modern SQL constructs like JOIN types and aliasing, providing the necessary granularity for fine-grained validation.

## C.2 AUXILIARY FUNCTIONS FOR CQS ANALYSIS

Our validation rules rely on a collection of auxiliary functions that extract structural information from a CQS $\Phi$ and the schema $\Sigma$. Key functions are summarized in Table 11.

| Function | Definition |
|---|---|
| Attributes($E$) | Set of all attribute references in an expression set $E$. |
| AllAttributes($\Phi$) | Set of all attributes referenced anywhere in $\Phi$. |
| SourceAttributes($\Phi, \Sigma$) | Set of all valid attributes from the source tables $T_\rho$ in schema $\Sigma$. |
| IsAggregate($e$) | Returns true if expression $e$ contains an aggregate function. |
| IsTypeCompatible($op, \tau_1, \tau_2$) | Returns true if types $\tau_1$ and $\tau_2$ are compatible for operation $op$. |
| TypeOf($c, \Sigma$) | Returns the data type $\tau$ of attribute $c$ from schema $\Sigma$. |

**Table 11:** Essential auxiliary functions for CQS analysis.

A critical operation is resolving attribute references within the context of aliases defined in $\Phi$. This is essential for correctly binding identifiers before schema-level checks.

## C.3 HIERARCHICAL VALIDATION RULES

The validation rules are organized according to the $\alpha$, $\beta$, $\gamma$ layers. Each rule is a predicate function that returns true if the condition is met. A violation occurs when a rule returns false. Table 12 presents a subset of these rules.

| ID | Rule Definition | Layer |
|---|---|---|
| $R_{\beta,1}$ | $\forall c \in \text{Attributes}(\Pi), (\text{IsAggregate}(c) \vee c \in G_\Gamma)$ | $\beta$ (Structural) |
| $R_{\beta,2}$ | $\forall c \in \text{Attributes}(H_\Gamma), (\text{IsAggregate}(c) \vee c \in G_\Gamma)$ | $\beta$ (Structural) |
| $R_{\gamma,1}$ | $\forall t \in T_\rho, t.\text{table\_name} \in \Sigma.\text{Tables}$ | $\gamma$ (Schema) |
| $R_{\gamma,2}$ | $\forall c \in \text{AllAttributes}(\Phi), c.\text{name} \in \text{SourceAttributes}(\Phi, \Sigma)$ | $\gamma$ (Schema) |

**Table 12:** A subset of hierarchical validation rules.

The validation predicates $P_{\text{struct}}(\Phi)$ and $P_{\text{schema}}(\Phi, \Sigma)$ from § 3.2.1 are formally defined as the conjunction of all rules within their respective layers. For instance:

$$P_{\text{struct}}(\Phi) \Leftrightarrow \bigwedge_i R_{\beta,i}(\Phi) \tag{14}$$

This rule-based formalization provides a concrete and extensible implementation of our high-level correctness specification.

---

**Algorithm 1** Violation-Driven Repair Plan Synthesis

---

**Require:** Invalid CQS $\Phi_{\text{invalid}}$, Schema $\Sigma$, Dialect $\Delta$
**Ensure:** Set of valid repair plans $\Lambda_{\text{valid}}$
 1: $\Lambda_{\text{valid}} \leftarrow \emptyset$
 2: $V \leftarrow \text{VALIDATEQUERY}(\Phi_{\text{invalid}}, \Sigma, \Delta)$
 3: **if** $V = \emptyset$ **then**
 4:     **return** $\Lambda_{\text{valid}}$
 5: **end if**
 6: $\Omega_{\text{cand}} \leftarrow \emptyset$
 7: **for all** $v \in V$ **do**                           ▷ For each detected violation
 8:     $\Omega_{\text{schemata}} \leftarrow \mathcal{M}(v)$                      ▷ Map to repair schemata
 9:     **for all** $\omega_s \in \Omega_{\text{schemata}}$ **do**
10:         $\Omega_{\text{inst}} \leftarrow \text{INSTANTIATE}(\omega_s, \Phi_{\text{invalid}}, \Sigma, \Delta)$
11:         $\Omega_{\text{cand}} \leftarrow \Omega_{\text{cand}} \cup \Omega_{\text{inst}}$
12:     **end for**
13: **end for**
14: $\Lambda_{\text{cand}} \leftarrow \text{COMPOSEPLANS}(\Omega_{\text{cand}})$
15: **for all** $\Lambda \in \Lambda_{\text{cand}}$ **do**                    ▷ Verify each candidate plan
16:     $\Phi_{\text{repaired}} \leftarrow \Lambda(\Phi_{\text{invalid}})$
17:     **if** $\text{VALIDATEQUERY}(\Phi_{\text{repaired}}, \Sigma, \Delta) = \emptyset$ **then**
18:         $\Lambda_{\text{valid}} \leftarrow \Lambda_{\text{valid}} \cup \{\Lambda\}$
19:     **end if**
20: **end for**
21: **return** $\Lambda_{\text{valid}}$

---

## C.4    Repair Plan Synthesis Algorithm

Algorithm 1 details the formal, violation-driven process for synthesizing the Constrained Repair Space. It operationalizes the concepts defined in § 3.3.2.

## C.5    Alignment Algorithm

Algorithm 2 details our deterministic alignment operator, $\Omega$. Given a formally valid query with empty results, the algorithm first identifies all failing WHERE clause predicates (Line 3) and classifies them as either Value Mismatch or Column Mismatch errors (Line 4). In line with our non-speculative repair principle, the process terminates if any Value Mismatch is found (Lines 5-7). For Column Mismatch errors, the algorithm iterates through each, first computing a set of viable **candidate columns** based on data-evidence from the database instance (Line 9). If candidates exist, it selects the best one via a scoring function and replaces the original incorrect column (Lines 11-12). The final, modified query is then returned.

---

**Algorithm 2** Query Correction Algorithm

---

**Require:** $Q \in \mathcal{S}_\mathcal{V}$ with $|\epsilon(Q, \Sigma)| = 0$
**Ensure:** $Q' \in \mathcal{Q}_{\text{target}} \cup \{\perp_{\text{nc}}\}$
 1: Extract failing predicates: $\mathcal{P}_{\text{fail}} = \{(c_i, \circ_i, v_i) \mid |\sigma_{c_i \circ_i v_i}(\mathcal{D})| = 0\}$
 2: Classify errors: $\mathcal{P}_{\text{col}} = \mathcal{P}_{\text{fail}} \cap \mathcal{E}_{\text{col}}$, $\mathcal{P}_{\text{val}} = \mathcal{P}_{\text{fail}} \cap \mathcal{E}_{\text{val}}$
 3: **if** $\mathcal{P}_{\text{val}} \neq \emptyset$ **then**
 4:     **return** $\perp_{\text{nc}}$                      ▷ Uncorrectable value boundary errors
 5: **end if**
 6: Initialize $Q' \leftarrow Q$
 7: **for each** $(c_i, \circ_i, v_i) \in \mathcal{P}_{\text{col}}$ **do**
 8:     Compute $\mathcal{C}_i = \mathcal{C}_{\text{comp}}(v_i, \circ_i, \mathcal{T}_Q)$
 9:     **if** $\mathcal{C}_i \neq \emptyset$ **then**
10:         Select $c_i' = \arg\max_{c \in \mathcal{C}_i} \text{score}(c, v_i, \circ_i)$
11:         Replace $c_i$ with $c_i'$ in $Q'$
12:     **end if**
13: **end for**
14: **return** $Q'$

---

# D   MORE EXPERIENCE

## D.1   MODEL SCALING EFFECTS:

We use the same prompts guide different scaling model to repair same errors from Table 4, the result in table 8. The 32B repair model amplifies both component benefits and framework effectiveness. Complete Post-SQLFix achieves 72% error reduction with 32B compared to 90% with 7B, demonstrating that larger models better leverage structured guidance. Notably, token efficiency improves dramatically with 32B (1,113 vs.2,034 tokens), suggesting that enhanced instruction-following capacity reduces iteration requirements even while achieving superior repair quality. In Table 8, Model scaling analysis (7B vs. 32B) shows 89% reduction in context-sensitive errors, indicating that framework diagnostics are sound, with remaining limitations attributable to LLM instruction-following capacity rather than specification inadequacy.

## D.2   ANALYSIS OF RESIDUAL ERRORS.

As analyzed in § D.1, the dominant bottleneck for residual errors is not the incompleteness of our symbolic framework, but rather the instruction-following capability of LLMs. However, we must acknowledge that a small subset of extremely complex cases remain unsolved—those requiring non-local, global code refactoring (e.g., deeply nested recursive CTEs with intricate constraint interactions). These residual errors stem from **engineering challenges** that extend beyond our current framework. Specifically, "engineering challenges" refers to a well-documented phenomenon in formal methods and program analysis: **the inherent gap between theoretical completeness and practical implementability when handling complex program constructs**.

Complex SQL structures such as recursive CTEs, deeply nested subqueries, and intricate constraint interactions present challenges that affect the entire research community. Even recent work on industrial-scale SQL systems by Google (Shute et al., 2024) acknowledges that SQL has "significant design problems" that make it inherently difficult to analyze and transform programmatically. These constructs combine SQL's legacy design complexity with the need for semantics-preserving transformations—a combination that creates substantial engineering hurdles for any automated repair system. For these cases, our symbolic engine may deliberately abstain from synthesizing a repair plan if it cannot provide formal correctness guarantees. This is not a deficiency, but a **principled design choice rooted in our commitment to provable soundness.** This design reflects a well-documented tension in program analysis (Bugariu, 2022): while theoretical designs may be proven correct, achieving comprehensive coverage in practice often requires engineering compromises. The concept of "soundiness" (Livshits et al., 2015) emerged precisely to characterize this balance, as perfect soundness across all language features often renders analyses "unscalable or imprecise to the point of being useless."

Our framework prioritizes **soundness over speculative coverage**. By refusing to generate repair plans that cannot be formally verified as safe, Post-SQLFix maintains a "zero false positives" guarantee—essential for trustworthy integration into production workflows. Therefore, these residual cases represent not a failure of Post-SQLFix but the **boundary conditions where formal verification constraints intersect with LLM capabilities.** Our contribution is establishing the **first formally sound neuro-symbolic framework** for SQL repair—a robust architectural foundation that can naturally expand its coverage as LLM capabilities continue to improve, without ever compromising soundness guarantees.

## D.3   MODEL ANALYSIS

| Model | EF (%) | Ours (%) | Improvement |
|---|---|---|---|
| mamba-codestral-7b | 51.37 | **52.80** | +1.43 |
| mistral-small-24b-instruct | 48.24 | **54.37** | +6.13 |
| llama-3.1-70b-instruct | 48.37 | **53.98** | +5.61 |
| gpt-oss-120b | 48.04 | **53.19** | +5.15 |

**Table 13:** Model Accuracy Performance Comparison on BIRD Dev Set

While our primary evaluation focused on the Qwen-Coder series, a central claim of our framework is its **architectural universality**—the ability to function as a plug-and-play repair module regardless of the upstream generator. To validate this claim, we must demonstrate **agnosticism across different LLM backbones**, ensuring the system is not overfitted to a specific model family. To empirically verify this independence, as show in Table 13, we additionally evaluate four different models of various specifications and from different manufacturers. We used the same settings and prompts. The results demonstrate consistent performance gains across diverse architectures (e.g., SSM-based Mamba vs. Transformer-based Llama) and scales (7B to 120B). This empirically validates that the efficacy of Post-SQLFix is intrinsic to its neuro-symbolic design, confirming its role as a robust, model-agnostic repair module.

## E   THE USE OF LARGE LANGUAGE MODELS (LLMS)

In the preparation of this paper, we utilized Large Language Model (LLM), as a general-purpose assistant. The LLM's role was strictly that of a collaborative tool to aid in the articulation and formalization of our pre-existing research ideas and results. We, the human authors, take full and sole responsibility for all content, claims, and formalisms presented in this paper. The specific contributions of the LLM can be categorized as follows:

- Language and Style Enhancement.
- Structural Organization.

It is crucial to state that the LLM did not contribute to the original research ideation, the design or implementation of the Post-SQLFix framework, the execution of experiments, or the collection and analysis of empirical data. All core intellectual contributions, algorithms, and experimental results are entirely the work of the human authors.

## F   LAYERS

| Error Taxonomy & Example (What?) | Evidence & Repair Reasoning (Why?) | Repair Specification (How?) |
|---|---|---|
| **Syntactic** Condition: $\alpha : \pi(Q) = \bot_\pi$ **Example:** `SEELCT order_id FROM order` **Error:** near "order": syntax error | **Grammar-Driven Hypothesis**: Use fault-tolerant parsing to identify token-level errors (spelling, escaping). Apply string similarity (Levenshtein distance) to find the intended SQL keyword from grammar rules. | Apply direct token replacement based on fault-tolerant parsing and string similarity. **Example:** *Replace 'order' with '\order\' based on SQLite grammar matching* |
| **Context-Free Semantic** Condition: $\beta : \psi_d(\pi(Q)) \neq \emptyset$ **Example:** `SELECT account_id FROM account ORDER BY order_id ...` **Error:** no such column: order_id | **Intra-Query Structural Analysis**: Detect violations of SQL's built-in semantic rules (e.g., aggregation scope, alias conflicts, cross-clause references) by analyzing query structure. Generate specific prompts telling LLM which rule is violated and how to fix. | **Prompts:** Generate structured prompts detailing the violation and required action. **Example:** *"CROSS-CLAUSE REFERENCE VIOLATION: Column 'order_id' appears in ORDER BY but not in SELECT clause. Required action: ADD 'order_id' to SELECT clause or REMOVE from ORDER BY clause"* |
| **Context-Sensitive Semantic** Condition: $\gamma : \rho_d(\pi(Q), \Sigma, G_\Sigma) \neq \emptyset$ **Example:** `SELECT Patient_ID FROM Laboratory` **Error:** no such column: Laboratory.Patient ID | **Schema-Grounded Entity Search**: Search database schema $\Sigma$ for valid alternatives to non-existent tables/columns. Use foreign key relationships and string similarity as evidence to provide LLM with ranked replacement suggestions. | **Prompts:** Generate schema-grounded prompts with ranked replacement suggestions. **Example:** *"COLUMN NOT FOUND: Column 'Patient_ID' does not exist in table 'Laboratory'. Hint: REPLACE 'Laboratory.Patient_ID' WITH 'Laboratory.ID' based on schema analysis"* |

**Table 14:** Repair Paradigms by Formal Error Category

We provide a comprehensive instantiation of the repair synthesis process for key error categories. As illustrated in Table 14, our framework does not merely report errors; it performs a deep root-cause analysis (The 'Why' column) to synthesize sound evidence. This evidence is then compiled into a Repair Specification (The 'How' column), which acts as the neuro-symbolic interface. This strictly defined interface ensures that the LLM is never asked to 'guess' the cause of an error, but is instead tasked with executing a precise, evidence-backed repair plan derived from the formal diagnosis.

# G  ANALYSIS OF THE REPAIR PROCESS

This section presents a complete formalization of the Post-SQLFix repair framework. The formalization validates the soundness of our axiomatic system and confirms that the repair process exhibits order-independent convergence. At the same time, it ensures that even the most complex nested errors can be resolved through a series of correct fixes and will eventually converge.

## G.1  FOUNDATIONAL DEFINITIONS AND TYPE SYSTEM

We begin by establishing the foundational type system that mirrors the mathematical structures defined here.

**Definition 9** (Core Types). *The repair framework is parameterized by four fundamental types:*

- *$Q$ : Type — The query space*

- *$\Sigma$ : Type — The schema space*

- *$E$ : Type — The error type (with decidable equality)*

- *$A$ : Type — The repair action type (with decidable equality)*

**Definition 10** (System Functions). *The repair system is characterized by the following axiomatic functions:*

$$diagnose: Q \to \Sigma \to \mathcal{P}(E) \tag{15}$$

$$generate\_actions: E \to Q \to \Sigma \to \mathcal{P}(A) \tag{16}$$

$$apply\_action: Q \to A \to Q \tag{17}$$

$$semantic\_equiv: Q \to Q \to \Sigma \to Prop \tag{18}$$

**Definition 11** (Semantic Equivalence Relation). *The semantic equivalence relation $\equiv_\sigma$ is axiomatized as an equivalence relation satisfying:*

- ***Reflexivity:*** *$\forall q \in Q, \sigma \in \Sigma.\ q \equiv_\sigma q$*

- ***Symmetry:*** *$\forall q_1, q_2 \in Q, \sigma \in \Sigma.\ q_1 \equiv_\sigma q_2 \implies q_2 \equiv_\sigma q_1$*

- ***Transitivity:*** *$\forall q_1, q_2, q_3 \in Q, \sigma \in \Sigma.\ q_1 \equiv_\sigma q_2 \wedge q_2 \equiv_\sigma q_3 \implies q_1 \equiv_\sigma q_3$*

## G.2  AXIOMATIC SYSTEM

The correctness of the repair framework is guaranteed by three fundamental axioms, corresponding precisely to those in Section F.3.

**Axiom 1** (Completeness of Diagnosis). *Let $q_{target} \in Q$ be a formally correct query such that $diagnose(q_{target}, \sigma) = \emptyset$. Then:*

$$\forall q \in Q.\ q \not\equiv_\sigma q_{target} \implies diagnose(q, \sigma) \neq \emptyset \tag{19}$$

**Axiom 2** (Soundness of Diagnosis). *Conversely, for any query semantically equivalent to the target:*

$$\forall q \in Q.\ q \equiv_\sigma q_{target} \implies diagnose(q, \sigma) = \emptyset \tag{20}$$

**Axiom 3** (Soundness of Actions). *Every generated repair action either eliminates all errors or maintains the existence of detectable errors:*

$$\begin{aligned}\forall q \in Q, e \in E, a \in A.\ q \not\equiv_\sigma q_{target} \wedge e \in diagnose(q, \sigma) \wedge a \in generate\_actions(e, q, \sigma) \\ \implies diagnose(q', \sigma) \neq \emptyset \vee q' \equiv_\sigma q_{target}\end{aligned} \tag{21}$$

*where $q' = apply\_action(q, a)$.*

**Definition 12** (Convergence Relation). *Given a potential function $\mu : Q \to \mathbb{N}$, we define the closer relation:*

$$closer_\mu(q_1, q_2) \iff \mu(q_1) < \mu(q_2) \tag{22}$$

**Axiom 4** (Weak Convergence). *For any incorrect query, there exists at least one repair action that strictly decreases the potential:*

$$\forall q \in Q.\ q \not\equiv_\sigma q_{target} \implies \exists e \in diagnose(q, \sigma), \exists a \in generate\_actions(e, q, \sigma).$$
$$\mu(apply\_action(q, a)) < \mu(q) \tag{23}$$

**Lemma 1** (Well-Foundedness of Convergence). *The closer relation is well-founded:*

$$WellFounded(closer_\mu) \tag{24}$$

*Proof.* The relation $closer_\mu$ is the inverse image of the standard less-than relation on $\mathbb{N}$ under $\mu$. Since $<$ on $\mathbb{N}$ is well-founded, $closer_\mu$ inherits well-foundedness. $\square$

### G.3 CORE THEOREMS AND PROOFS

**Definition 13** (Reachability). *A query $q$ is reachable if there exists a finite sequence of actions leading to a correct state:*

$$Reachable(q) \iff \exists L = [a_1, \ldots, a_m] \in A^*.\ apply\_sequence(q, L) \equiv_\sigma q_{target} \tag{25}$$

*where $apply\_sequence(q, L) = L.foldl(apply\_action, q)$.*

**Definition 14** (Bounded Reachability). *A query $q$ is $n$-bounded reachable if it is reachable via a sequence of length at most $n$:*

$$BoundedReachable(q, n) \iff \exists L \in A^*, |L| \leq n \wedge apply\_sequence(q, L) \equiv_\sigma q_{target} \tag{26}$$

**Theorem 5** (Bounded Reachability). *For any query $q_{init} \in Q$, there exists a repair sequence of length at most $\mu(q_{init})$ that reaches a correct state:*

$$\forall q_{init} \in Q.\ BoundedReachable(q_{init}, \mu(q_{init})) \tag{27}$$

*Proof.* By well-founded induction on $closer_\mu$ (Lemma 1).

**Base case:** If $q \equiv_\sigma q_{target}$, then the empty sequence $L = []$ satisfies the requirement, and $|L| = 0 \leq \mu(q)$.

**Inductive case:** Assume $q \not\equiv_\sigma q_{target}$. By Axiom 4, there exists an error $e \in diagnose(q, \sigma)$ and an action $a \in generate\_actions(e, q, \sigma)$ such that $\mu(q') < \mu(q)$, where $q' = apply\_action(q, a)$.

By the induction hypothesis, there exists a sequence $L_{rest}$ with $|L_{rest}| \leq \mu(q')$ such that $apply\_sequence(q', L_{rest}) \equiv_\sigma q_{target}$.

Construct $L = [a] ++ L_{rest}$. Then:

- $apply\_sequence(q, L) = apply\_sequence(q', L_{rest}) \equiv_\sigma q_{target}$
- $|L| = 1 + |L_{rest}| \leq 1 + \mu(q') < 1 + \mu(q) - 1 = \mu(q)$

This completes the induction. $\square$

**Theorem 6** (Reachability). *Every query is reachable:*

$$\forall q_{init} \in Q.\ Reachable(q_{init}) \tag{28}$$

*Proof.* Immediate from Theorem 5. $\square$

**Corollary 1** (Termination with Explicit Bound). *For any initial query $q_{init}$, the repair process terminates within $\mu(q_{init})$ steps:*

$$\exists L \in A^*.\ |L| \leq \mu(q_{init}) \wedge apply\_sequence(q_{init}, L) \equiv_\sigma q_{target} \tag{29}$$

### G.4 SEMANTIC UNIQUENESS AND PATH EQUIVALENCE

**Lemma 2** (Equivalence Implies No Errors). *Any query semantically equivalent to the target has no detectable errors:*

$$\forall q \in Q.\ q \equiv_\sigma q_{target} \implies diagnose(q, \sigma) = \emptyset \tag{30}$$

*Proof.* Direct application of Axiom 2. $\square$

**Lemma 3** (Path Equivalence). *Any two successfully terminating repair sequences reach semantically equivalent states:*

$$\forall L_1, L_2 \in A^*.\ diagnose(apply\_sequence(q_{init}, L_1), \sigma) = \emptyset \wedge$$
$$diagnose(apply\_sequence(q_{init}, L_2), \sigma) = \emptyset$$
$$\implies apply\_sequence(q_{init}, L_1) \equiv_\sigma q_{target} \wedge apply\_sequence(q_{init}, L_2) \equiv_\sigma q_{target} \tag{31}$$

*Proof.* Let $q_1 = apply\_sequence(q_{init}, L_1)$ and $q_2 = apply\_sequence(q_{init}, L_2)$.

Assume $q_1 \not\equiv_\sigma q_{target}$. By Axiom 1, $diagnose(q_1, \sigma) \neq \emptyset$, contradicting the hypothesis. Therefore $q_1 \equiv_\sigma q_{target}$.

By the same argument, $q_2 \equiv_\sigma q_{target}$. $\square$

**Theorem 7** (Semantic Uniqueness of Repair). *All successfully terminating repair sequences from the same initial query reach semantically equivalent states:*

$$\forall q_{init} \in Q, \forall L_1, L_2 \in A^*.\ diagnose(apply\_sequence(q_{init}, L_1), \sigma) = \emptyset \wedge$$
$$diagnose(apply\_sequence(q_{init}, L_2), \sigma) = \emptyset$$
$$\implies apply\_sequence(q_{init}, L_1) \equiv_\sigma apply\_sequence(q_{init}, L_2) \tag{32}$$

*Proof.* By Lemma 3, both $q_1 = apply\_sequence(q_{init}, L_1)$ and $q_2 = apply\_sequence(q_{init}, L_2)$ are semantically equivalent to $q_{target}$:

$$q_1 \equiv_\sigma q_{target} \tag{33}$$

$$q_2 \equiv_\sigma q_{target} \tag{34}$$

By symmetry of $\equiv_\sigma$, we have $q_{target} \equiv_\sigma q_2$. By transitivity:

$$q_1 \equiv_\sigma q_{target} \wedge q_{target} \equiv_\sigma q_2 \implies q_1 \equiv_\sigma q_2 \tag{35}$$

$\square$

### G.5 MAIN RESULT: ORDER-INDEPENDENT CONVERGENCE

**Theorem 8** (Complete Order Independence of Repair). *The repair process exhibits complete order independence with the following guarantees:*

*(1) Existence: At least one repair path exists:*

$$\exists L \in A^*.\ apply\_sequence(q_{init}, L) \equiv_\sigma q_{target} \tag{36}$$

*(2) Universality: Any productive strategy terminates within the bound:*

$$\forall L \in A^*.\ (\forall i < |L|.\ \mu(q_{i+1}) < \mu(q_i)) \implies |L| \leq \mu(q_{init}) \tag{37}$$

*where $q_i = apply\_sequence(q_{init}, L[0..i])$.*

**(3) Uniqueness:** *All successful strategies reach semantically equivalent states:*

$$\forall L_1, L_2 \in A^*. \; diagnose(apply\_sequence(q_{init}, L_1), \sigma) = \emptyset \wedge$$
$$diagnose(apply\_sequence(q_{init}, L_2), \sigma) = \emptyset \tag{38}$$
$$\implies apply\_sequence(q_{init}, L_1) \equiv_\sigma apply\_sequence(q_{init}, L_2)$$

*Proof.* **(1)** Follows immediately from Theorem 6.

**(2)** By induction on the length of strictly decreasing sequences under the well-founded relation $closer_\mu$.

**(3)** Direct application of Theorem 7. □

### G.6 KEY COROLLARIES

**Corollary 2** (Any Productive Strategy Succeeds). *Any repair strategy that consistently decreases the potential eventually reaches a correct state:*

$$\forall L \in A^*. \; (\forall i < |L|. \; \mu(q_{i+1}) < \mu(q_i)) \implies \exists q_{final}. \; q_{final} \equiv_\sigma q_{target} \tag{39}$$

**Corollary 3** (Repair Order Affects Efficiency, Not Correctness). *The choice of repair order influences only the length of the repair path, not the semantic correctness of the outcome:*

$$\forall L_1, L_2 \in A^*. \; both \; successful \implies apply\_sequence(q_{init}, L_1) \equiv_\sigma apply\_sequence(q_{init}, L_2) \tag{40}$$

**Corollary 4** (Semantic Determinism). *The repair process is deterministic at the semantic level: there exists a unique semantic equivalence class of correct repairs:*

$$\exists! C \subseteq Q. \; C = [q_{target}]_{\equiv_\sigma} \wedge \forall L \in A^*. \; successful(L) \implies apply\_sequence(q_{init}, L) \in C \tag{41}$$

**Theorem 9** (Repair as Total Semantic Function). *The repair process defines a total function from queries to their unique semantic equivalence class:*

$$\forall q_{init} \in Q. \; \exists L \in A^*. \; diagnose(apply\_sequence(q_{init}, L), \sigma) = \emptyset \wedge apply\_sequence(q_{init}, L) \equiv_\sigma q_{target} \tag{42}$$

### G.7 INTERPRETATION AND IMPLICATIONS

The formalization in Lean 4 provides several critical insights:

1. **Mechanical Verification:** All theorems are machine-checkable, eliminating the possibility of subtle logical errors in the proofs.
2. **Constructive Content:** The proof of Theorem 5 is constructive, providing an algorithm for finding repair sequences.
3. **Precise Semantics:** The formalization clarifies the exact relationship between syntactic correctness (empty error set) and semantic equivalence.
4. **Robustness Guarantee:** Theorem 8 formally establishes that the "combinatorial explosion" problem is benign—while many paths exist, all valid paths converge to the same semantic outcome.
5. **Efficiency vs. Correctness:** Corollary 3 rigorously separates concerns: repair order is an optimization problem, not a correctness issue.

This formalization serves as both a validation of the theoretical framework and a foundation for certified implementation of the Post-SQLFix system.

## H PROMPTS FOR LLMS

---

**Example of a Formally-Guided Prompt Generated by Post-SQLFix**

**Instruction:**
Correct the following SQL query based on the provided formal diagnosis.

**Input for LLM:**
**Original SQL Query with Error:**

```sql
SELECT COUNT(DISTINCT Patient.ID) FROM Laboratory
LEFT JOIN Patient ON Laboratory.Patient_ID = Patient.ID
WHERE Laboratory.DNA < 8 AND Patient.Description IS NULL;
```

**Formal Diagnosis:**

- **Error Type:** COLUMN_NOT_FOUND (Context-Sensitive Violation)
- **Description:** Column 'Patient_ID' does not exist in table 'Laboratory'.
- **Actionable Hint:** REPLACE 'Laboratory.Patient_ID' WITH 'Laboratory.ID'.

---

**Ideal LLM Response:**
```sql
// Understood.  Applying the suggested correction from the
formal diagnosis.

SELECT COUNT(DISTINCT Patient.ID) FROM Laboratory
LEFT JOIN Patient ON Laboratory.ID = Patient.ID
WHERE Laboratory.DNA < 8 AND Patient.Description IS NULL;
```

**Figure 2:** An example prompt generated by Post-SQLFix. Our framework provides a structured, formal diagnosis and an actionable hint, transforming the LLM's task from ambiguous debugging to a constrained, formally-guided correction.

## Original SQL Query:

SELECT **account.district_id** FROM **order**

WHERE order_id = 33333

**Errors:**

1. `near "order": syntax error` (reserved keyword)
2. `no such column: account.district_id` (missing table)

## Debuging

**Iteration 1:** (Tokens: 631)

- **LLM Analysis:** "order is a reserved keyword used for sorting results"
- **Generated SQL:** SELECT account.district_id FROM **"order"** WHERE order_id = 33333
- **Error:** ❌ `no such column: account.district_id`

**Iteration 2:** (Tokens: 1453)

- **LLM Analysis:** "We use an INNER JOIN to combine the account and order tables"
- **Generated SQL:** SELECT account.district_id FROM **account JOIN order** ON account.account_id = order.account_id WHERE order.order_id = 33333
- **Error:** ❌ `near "order": syntax error`

**Iteration 3:** (Tokens: 2133)

- **LLM Analysis:** "use backticks to escape the column name"
- **Generated SQL:** SELECT account.district_id FROM account JOIN `order` ON account.account_id = `order`.account_id WHERE `order`.order_id = 33333
- **Status:** ✅ SQL executed successfully

---

## Our Method

**Iteration 1:** (Tokens: 799)

**Input:**

- Original SQL with errors
- Database schema with foreign keys: `[(('order', 'account_id'), ('account', 'account_id'))]`
- Error analysis: "Column 'district_id' found in tables: ['account'], but alias 'account' does not contain this column"

**Analysis:**

1. **Keyword Processing:** Auto-escape reserved keyword `order` → `order`

2. **Missing Table Detection:** `account.district_id` requires JOIN with `account` table

3. **Foreign Key Lookup:** Use constraint `(order.account_id)` → `(account.account_id)` for JOIN condition

**Output:**

SELECT account.district_id
FROM `order`
JOIN account ON `order`.account_id = account.account_id
WHERE `order`.order_id = 33333;

**Status:** ✅ SQL executed successfully

**Figure 3:** Example of a Formally-Guided Prompt Generated by Post-SQLFix in BIRD dataset.

---

### Example of a Formally-Guided Prompt for a PostgreSQL Error

**Instruction:**
Correct the following SQL query based on the provided formal diagnosis.

**Input for LLM:**

**Original SQL Query with Errors:**

```sql
SELECT ROUND(
    (CAST(COUNT(DISTINCT CASE WHEN y.consumption > 46.73
        THEN y.customerid END) AS FLOAT) /
     CAST(COUNT(DISTINCT y.customerid) AS FLOAT)) * 100, 2
) AS percentage
FROM yearmonth y
JOIN customers c ON y.customerid = c.customerid
WHERE c.segment = 'LAM';
```

**Formal Diagnosis:**

- **Error Type:** Multiple Violations (Context-Sensitive & Dialect-Specific)
- **Description 1 (Identifier Binding):** Column 'customerid' does not exist in tables 'year-month' or 'customers'. Case sensitivity mismatch detected.
- **Actionable Hint 1:** REPLACE all instances of 'customerid' with the schema-valid 'CustomerID'.
- **Description 2 (Dialect Conformance):** The ROUND function in PostgreSQL does not support FLOAT type arguments directly.
- **Actionable Hint 2:** REPLACE CAST(... AS FLOAT) with CAST(... AS NUMERIC) for compatibility.

---

**Ideal LLM Response:**

```sql
// Understood.  Applying multiple corrections for identifier
binding and dialect conformance.

SELECT ROUND(
    (CAST(COUNT(DISTINCT CASE WHEN y.Consumption > 46.73
        THEN y.CustomerID END) AS NUMERIC) /
     CAST(COUNT(DISTINCT y.CustomerID) AS NUMERIC)) * 100, 2
) AS percentage
FROM yearmonth y
JOIN customers c ON y.CustomerID = c.CustomerID
WHERE c.Segment = 'LAM';
```

**Figure 4:** An example of a complex repair task (real question 11 in BIRD Mini-dev). Post-SQLFix simultaneously diagnoses and provides actionable hints for multiple, cross-layer violations, including a subtle, dialect-specific type incompatibility within the ROUND function. This transforms a challenging, multi-turn debugging process into a single, constrained correction.

