# OpenReview forum: "From Ambiguous Feedback to Verifiable Repair via Formal Synthesis in Text-to-SQL"
_ICLR.cc/2026/Conference — Submitted to ICLR 2026_

### Official Review · Reviewer_vH8N · 2025-10-29

**Soundness:** 3
**Presentation:** 2
**Contribution:** 3
**Rating:** 6
**Confidence:** 4

**Summary:**

This paper proposes a neurosymbolic repair for SQL that is applied to incorrect SQL
generated from NL using LLMs.

The popular approach here is self-debugging, where if the generated SQL throws error
on execution, the errors are presented to the LLM and the LLM is asked to repair its
incorrect generation conditioned on the errors. This "execution feedback" (EF) does not
guarantee that errors will be fixed, and it could get into multistep feedback loops.

In this work, the authors build a symbolic engine that analyzes an incorrect SQL
and identifies the violations. These violations are then mapped to potential repairs
(schemas of repairs, not the concrete repair). The LLM is now presented the findings
of the symbolic analyzer, and the LLM suggests repairs, and the one that does not throw
any errors on execution is finally accepted. This post processor is called Post-SQLFix.

The paper shows that Post-SQLFix performs better than EF and even uses fewer iterations
than EF. The paper also shows that Post-SQLFix can also be used to improve SQL generation
from any model since it can be plugged to the output of any model. This is shown on two
datasets, BIRD and Spider.

**Strengths:**

Strengths:
1. The idea of using a symbolic analyzer to guide the repair is a powerful paradigm,
and the authors successfully demonstrated that it works for Text-to-SQL.
2. The experiments are fairly exhaustive and provide good evidence for the main claims
of the paper.

**Weaknesses:**

Weaknesses:
1. The writing is rushed, which makes the paper a confusing read at many places.
2. The symbolic parts of the Post-SQLFix are not entirely trivial and may have
required a lot of effort. Most of the insights are specific to SQL.
Question: How much work was it to get all the symbolic pieces together, and what learnings could
one take from here when working on another domain?

Details:
1. Line 262-263 says "we provide a detailed instantiation of this mapping for key violation
types in Appendix C.4" -- but there is no such thing in Appendix C.4
2. Section 3.3.2 is really hard to follow due to a lack of details. Who composes the violations
into candidate repair plans? Is it a symbolic procedure or LLM? If it is an LLM, then why is
it a repair plan and not just a repair?
3. Theorem 1 mentions "our synthesis function" -- it is not clear what this is. Is this Algorithm 1?
But Algorithm 1 is not called "synthesis". Please try to be rigorous when writing
statements of theorems. In the next section, "Synthesize" function is not even generating a CQS.
4. Definition 5: Last line, should = be op? If not, then what is the role of "op" ?
5. Table 5, does bold mean "better"? If so, then Time for EF is better than time for "Our" and yet
time for "Our" is in bold.
6. The distinction between "Universal Efficacy and Integration" and "Plug-and-Play" is unclear,
and I can't even guess what the difference might be in this case. Line 402 "By intercepting and resolving
errors early" hints that I am missing some important detail.
7. Line 400 mentions 54.9% vs 53.1%  -- is this statistically significant? What was the variance
observed in the 5 runs?
8. Line 460: "the limitation of EF is architectural, not informational": what does this mean?

The authors may wish to see some related work in [Poesia ICLR 2022], [Bavishi OOPSLA 2022],
and https://www.arxiv.org/abs/2508.09324  that all share some of the same high-level philosophy
as this work.

Other typos/errors:
l26: EF in abstract is undefined
l67: "correctness-intent tradeoff" : Why would this be a trade-off, they are not mutually
adversarial.
l130: "In contrast,regardless of their architectural complexity, converge on a different
fundamental dependency" -- What does that mean?
l229: Equation (3) - Is the negation needed there?
l262: "a Algorithm 1 first identifies the set of violations V" -- incomplete sentence.
l266: "Only those plans produce a verifiably valid successor state are included in the final output." --
unclear what this means.
l269: "for any error our identifies" -- missing word.
l320: "We seeks to ..."

**Questions:**

See the Weaknesses section above.

---

> ### Author Response · Authors · 2025-11-23
> **Part 1**
>
> We sincerely thank the reviewer for their meticulous evaluation and for **recognizing our neuro-symbolic approach as a "powerful paradigm" supported by "fairly exhaustive" experiments**. We are particularly grateful for your sharp eye in identifying the referencing inconsistencies and typographical errors (Q1, Q4, Q5). Your attention to detail has significantly helped us improve the manuscript's precision. In the response below, we confirm that all identified errors have been corrected, and we provide further clarifications on the symbolic engine’s mechanism (Q2, Q3) and the architectural distinctions (Q6, Q8) of our framework.
>
> **Q1:** Line 262-263 says "we provide a detailed instantiation of this mapping for key violation types in Appendix C.4" -- but there is no such thing in Appendix C.4
>
> The detailed examples illustrating the mapping from violations to repair plans were intended to be presented in **Table 13** on page 20. We have corrected the reference in the future revised manuscript to point to Table 13.
>
> **Q2:** Section 3.3.2 is really hard to follow due to a lack of details. Who composes the violations into candidate repair plans? Is it a symbolic procedure or LLM? If it is an LLM, then why is it a repair plan and not just a repair?
>
> The entire process described in $\S$ 3.3.2 does not involve LLMs.  The identifying violations, retrieving operator schemata via the strategic mapping $M$, instantiating them, and composing them into candidate repair plans, is all performed by our symbolic engine.
>
> A "repair plan" is a formal sequence of our defined "repair operators" (Definition 1 & 2); it is a **specification for a repair**, not the final repaired query itself. The composition (Algorithm 1) is a symbolic procedure for assembling these operators. We will revise $\S$ 3.3.2 and the description of Algorithm 1 to make this distinction crystal clear, emphasizing that this is a purely symbolic synthesis process preceding any interaction with an LLM.
>
> **Q3:** Theorem 1 mentions "our synthesis function" -- it is not clear what this is... In the next section,  the "Synthesize" function is not even generating a CQS.
> Yes, the synthesis function does not output CQS.
> *   The synthesis function in **Theorem 1** formally refers to the procedure encapsulated in **Algorithm 1**, which we should have named `Synthesize_Plans`. Its output is a set of valid Repair Plans.
> *   The function `Synthesize_spec` in **Definition 3** is a higher-level function representing the entire symbolic engine's public API. It internally calls `Synthesize_Plans` (Algorithm 1) and then formats the output into the final Repair `Specification` ($S_R$).
>
> In future revise manuscript, we have renamed the function in Algorithm 1 to `Synthesize_Plans` and added a sentence directly before Theorem 1 clarifying that the theorem refers to this specific algorithm.
>
> **Q4:** Definition 5: Last line, should $=$ be $op$? If not, then what is the role of "op" ?
>
> The corrected line should read: $\exists$ tuple $\in$ $D$ such that $tuple[c*]$ $op$ $v$ This ensures we are seeking a column $c*$ that makes the original logical condition true for at least one row. We have corrected this in Definition 5.
>
> **Q5:** Table 5, does bold mean "better"? If so, then Time for EF is better than time for "Our" and yet time for "Our" is in bold.
>
> Yes, bold number means "better". Thanks for catching the typo. After checking the `Time (s)` in table, our data is written in the opposite order to EF. We have corrected the number and bolding in Table 5.
>
> **Q6:** The distinction between "Universal Efficacy and Integration" and "Plug-and-Play" is unclear, and I can't even guess what the difference might be in this case. Line 402 "By intercepting and resolving errors early" hints that I am missing some important detail.
>
> These two modes refer to two different ways our framework can be deployed:
> 1.  **Plug-and-Play(Tables 4,5,6,9):** This refers to using Post-SQLFix as a post-hoc correction module. It takes a *final, complete but incorrect SQL query* from any existing Text-to-SQL model and attempts to fix it. This demonstrates its utility as a standalone tool that can be used to improve existing systems without any changes.
> 2.  **Universal Efficacy and Integration(Tables 2 & 3):** This refers to a scenario where Post-SQLFix is deeply integrated into the generation pipeline itself. The formalizability of database schemas enables our system to capture and rectify any database-related errors in LLM outputs at the earliest possible stage, before propagation to downstream components. The phrase "intercepting and resolving errors early" refers to this mode. By catching an error in an intermediate step, we prevent it from propagating. This early intervention is why this mode achieves the highest performance gains.
>
> We will revise the text in $\S$ 4.2.1 to make this distinction much clearer.

---

> ### Author Response · Authors · 2025-11-23
> **Part 2**
>
> **Q7:** Line 400 mentions 54.9% vs 53.1% -- is this statistically significant? What was the variance observed in the 5 runs?
>
> Yes, the standard deviation for both results across the 5 runs was very low. For the Gen-SQL model on the BIRD dataset, the mean accuracies were 54.9% with a standard deviation of 0.9% and 53.1% with a standard deviation of 0.7%.
>
> **Q8:** Line 460: "the limitation of EF is architectural, not informational": what does this mean?
>
> The core limitation with the execution feedback (EF) paradigm is not the *quantity* of information in an error message. The limitation is the **architecture** that processes this information. EF's Architecture is to force a probabilistic LLM to act as a debugger, attempting to infer a root cause from a textual *symptom*. This is fundamentally unreliable. Our architecture introduces a symbolic engine designed for **root-cause diagnosis**. It translates the symptom into a formal, structured diagnosis and a set of verifiable repair plans.
>
> **Q9**: The symbolic parts of the Post-SQLFix are not entirely trivial and may have required a lot of effort. Most of the insights are specific to SQL. Question: How much work was it to get all the symbolic pieces together, and what learnings could one take from here when working on another domain? The authors may wish to see some related work in [Poesia ICLR 2022], [Bavishi OOPSLA 2022], and https://www.arxiv.org/abs/2508.09324 that all share some of the same high-level philosophy as this work.
>
> While Post-SQLFix's symbolic component required substantial implementation (>8K line of code and >500 testcases), the **architectural design, not the code volume, constitutes our core contribution**.
>
> Many existing approaches focus on constrained decoding during code generation. Systems like Synchromesh [1] and subsequent work on type-constrained decoding [3,4] ensure formal correctness by restricting the LLM's choices at the token level during generation. While highly effective for ensuring syntactic and type correctness, this generation-time approach is fundamentally ill-suited for repairing complex, non-local semantic errors that require a holistic understanding of the entire program structure. **These existing methods excel at preventing malformed syntax but cannot address errors whose root causes are distant from their manifestation points, a well-known challenge in program debugging [6]**.
> Other neurosymbolic approaches employ SMT solvers for formal verification [2,5]. While SMT solvers provide rigorous logical guarantees, translating their satisfying assignments—often consisting of abstract constraint solutions—into meaningful, intent-preserving code repairs remains a process of manual, fragile, and often complex prompt engineering. The gap between formal satisfiability and practical code repair represents a significant barrier to automation.
> Our approach is fundamentally different. **Post-SQLFix performs a holistic, post-hoc diagnosis to understand the root cause of all errors, then automatically compiles this structural understanding into an actionable Repair Specification ($S_R$) that guides LLM repair without manual prompt engineering.**
>
> For other domains, our key insights are:
> 1. Separation of concerns: decouple deterministic verification from probabilistic repair generation.
> 2. Hierarchical validation: multi-layer checking enables precise error localization and targeted repair.
>
> **Any domain requiring reliable LLM-generated code with formal correctness guarantees can benefit from our neuro-symbolic paradigm.** Success depends primarily on designing appropriate abstractions for the target domain's error taxonomy and repair semantics.
>
> We will expand our related work to include detailed comparisons with the suggested references. Thank you again!
>
> **Refernce**:
> 1. Synchromesh: Reliable code generation from pre-trained language models (ICLR'22)  https://arxiv.org/abs/2201.11227v1
> 2. Neurosymbolic Repair for Low-Code Formula Languages (OOPSLA'22) https://dl.acm.org/doi/10.1145/3563350
> 3. Type-Constrained Code Generation with Language Models (PLDI'25) https://dl.acm.org/doi/epdf/10.1145/3729274
> 4. Correctness-Guaranteed Code Generation via Constrained Decoding (COLM'25) https://arxiv.org/abs/2508.15866
> 5. SMT Solver Validation Empowered by LLMs (ASE'23) https://dlnext.acm.org/doi/10.1109/ASE56229.2023.00180
> 6. A study of the effectiveness of dynamic slicing in locating real faults (Empirical Software Engineering'07) https://doi.org/10.1007/s10664-006-9007-3

---

### Official Review · Reviewer_4g3n · 2025-10-30

**Soundness:** 3
**Presentation:** 3
**Contribution:** 3
**Rating:** 4
**Confidence:** 4

**Summary:**

This paper targets the problem that SQL queries generated by LLMs often contain formal errors and rely on vague "Execution Feedback" (EF) for repairs. It proposes the Post-SQLFix neuro-symbolic framework. This method uses a "Canonical Query Structure" (CQS) to perform a formal diagnosis of the query and employs "constrained synthesis" to generate a guaranteed-correct repair plan. This transforms the LLM from a guesser into a constrained executor, significantly improving accuracy by up to 11.6% on BIRD and Spider and reducing repair iterations by 50%.

**Strengths:**

1. The paper's greatest contribution is proposing a new paradigm for Text-to-SQL repair. It successfully transforms the uncertain, LLM-guess-dependent "execution feedback" loop into a deterministic, formal "diagnose-synthesize-execute" workflow.
2. The core innovation lies in the "constrained synthesis" of a guaranteed-valid repair plan space. This fundamentally changes the LLM's role in SQL repair and reduces the risk of introducing new errors during the process.
3. The framework's improvements in accuracy and reduction in iteration costs are quite significant. In particular, the results on multi-dialect benchmarks demonstrate that this method, when handling dialect-specific errors, is superior to EF methods that rely on the LLM to interpret different error messages.

**Weaknesses:**

1. The framework only guarantees "formal correctness," not "intent correctness." A SQL query might fully comply with syntax, type, and schema constraints but be completely misaligned with the user's intent due to faulty join logic. Post-SQLFix does not seem capable of detecting or fixing such errors.
2. The paper also mentions that residual errors mainly come from "complex SQL structures" (e.g., deep nesting, recursion) because their repair logic is "combinatorially explosive." This indicates that "constrained synthesis" faces complexity challenges in practice, and its "repair completeness" is bounded.
3. The paper's writing when introducing the overall workflow is somewhat overly obscure, with a suspicion of intentionally piling on formulas.

**Questions:**

1. There are actually many trained SQL generation models for the Bird-SQL dataset. Do these types of models exhibit a phenomenon where formal correctness capability increases while intent correctness capability decreases? In such cases, would the Post-SQLFix method still be effective? The paper only uses two sizes of the Qwen-coder model, making it somewhat difficult to prove the method's generalizability.
2. The paper mentions the "combinatorial explosion" problem for complex repairs. Does this mean the method is powerless against errors in cases like multi-level nesting or complex functions? Can you analyze where the practical capability boundary of "constrained synthesis" lies?
3. When handling empty results, why not continue using the neuro-symbolic collaborative approach (e.g., having the symbolic engine generate candidate repairs and then having the LLM judge the intent)? Why does it instead revert to a purely deterministic, restricted rule?

---

> ### Author Response · Authors · 2025-11-23
> **Part 1**
>
> We thank the reviewer for high evaluation of our work, particularly for **recognizing Post-SQLFix as a "new paradigm" that successfully transforms the uncertain execution feedback loop into a deterministic "diagnose-synthesize-execute" workflow**. We are deeply encouraged by your insight that **our core innovation constrained synthesis fundamentally redefines the LLM's role to reduce error risks**, and we value your validation of **our framework's superiority on multi-dialect benchmarks**. In the response below, we address your constructive concerns regarding the boundary between formal and intent correctness (W1), the accessibility of our formalisms (W3), and the system's robustness under complexity (Q2), while providing new experimental evidence of our method's generalizability (Q1).
>
> **W1:** The framework only guarantees "formal correctness," not "intent correctness." A SQL query might fully comply with syntax, type, and schema constraints but be completely misaligned with the user's intent due to faulty join logic. Post-SQLFix does not seem capable of detecting or fixing such errors.
>
> Yes, Post-SQLFix is explicitly designed to guarantee **formal correctness**, not intent correctness. This is a deliberate scope choice driven by the principle of zero false positives. Formal correctness is the non-negotiable prerequisite. A query that cannot execute (due to syntax or schema errors) is useless, regardless of its intent. **Our goal is to solve this foundational layer perfectly. By replacing unreliable execution feedback with a deterministic engine, we ensure the SQL is valid and runnable.**
>
> **Architectural Positioning (Plug-and-Play):** Post-SQLFix is designed as a deterministic backend module. Intent alignment is an orthogonal task best handled by a specialized frontend (e.g., a reasoning agent or a separate research module). **Post-SQLFix's Role is to act as the final safety guard. Users can plug Post-SQLFix after any front-end that focuses on specific functions to ensure that its output conforms to the database schema**.
>
>
> **W3:** The paper's writing when introducing the overall workflow is somewhat overly obscure, with a suspicion of intentionally piling on formulas.
>
> We appreciate this feedback on readability. Our use of formal notation allows us to mathematically prove that our repair plans are valid (Theorem 1) and that the process converges (Theorem 8), differentiating our work from EF-based heuristics. While the formal foundation is essential for our guarantees, we commit to improving accessibility in the revision. We will add "pedagogical scaffolding"—intuitive, plain-text explanations accompanying each definition to bridge the gap between the high-level workflow and the formal proofs.
>
> **Q1:** There are actually many trained SQL generation models for the Bird-SQL dataset. Do these types of models exhibit a phenomenon where formal correctness capability increases while intent correctness capability decreases? In such cases, would the Post-SQLFix method still be effective? The paper only uses two sizes of the Qwen-coder model, making it somewhat difficult to prove the method's generalizability.
>
> We tested the hypothesis by extending our evaluation to four additional diverse LLMs, including both specialized code and general chat models. The results (Table R1) contradict the "trade-off" hypothesis. Post-SQLFix consistently improves execution accuracy across all model types.
>
> **Table R1: Performance on Additional LLMs (BIRD Dev Set)**
> | Model | EF (%) | Post-SQLFix (%) | Improvement |
> | :--- | :---: | :---: | :---: |
> | mamba-codestral-7b | 51.37 | 52.80 | +1.43 |
> | mistral-small-24b  | 48.24 | 54.37 | +6.13 |
> | llama-3.1-70b | 48.37 | 53.98 | +5.61 |
> | gpt-oss-120b | 48.04 | 53.19 | +5.15 |
>
> **Safeguarding Intent**: The reason we do not see a decrease in intent correctness is architectural. To quantify this, we measured Non-Compliance Errors (where the model deviates from the plan, potentially altering intent). As shown in Table R2, such deviations are rare, and crucially, our validation loop (Figure 1(b)) captures 100% of them. This ensures that formal correctness is never achieved at the expense of semantic fidelity.
>
> **Table R2: Robustness to Instruction Non-Compliance**
> | Model | Non-Compliant Errors | Captured by Validator |
> | :--- | :---: | :---: |
> | Qwen2.5-7B | 2 | 100% |
> | Qwen2.5-32B | 0 | N/A |
> | mamba-codestral-7B | 2 | 100% |
> | mistral-small-24b | 1 | 100% |
> | llama-3.1-70b | 15 | 100% |
> | gpt-oss--120B | 15 | 100% |
>
> **Generalizability**: Our framework is architecturally model-agnostic. By decoupling formal verification (Symbolic Engine) from intent generation (LLM), we allow base models to be optimized purely for intent alignment. **Post-SQLFix then serves as a universal "Plug-and-Play" backend to guarantee executability, making it more valuable in a diverse ecosystem of Chat and Code models.**

---

> ### Author Response · Authors · 2025-11-23
> **Part 2**
>
> **Q2 & W2:** The paper mentions the "combinatorial explosion" problem for complex repairs. Does this mean the method is powerless against errors in cases like multi-level nesting or complex functions? Can you analyze where the practical capability boundary of "constrained synthesis" lies?
>
> Post-SQLFix can repair errors in multi-level nesting or complex functions cases. As detailed in our response to Reviewer XYmR (Q3), we resolve the "combinatorial explosion" through three mechanisms:
> - **Aggressive Pruning***: Our hierarchical validation and schema-based constraints prune the search space early, preventing exponential branching (detailed in ).
> - **Theoretical Convergence**: We prove Order-Independent Reachability (future revised paper Theorem 8). This guarantees that finding any valid repair step is sufficient to converge to a correct state, eliminating the need to exhaustively search all combinatorial paths.
> - **Constrained Execution**: The LLM is presented with a finite, verifiable candidate set, transforming it from an open-ended generator into a constrained selector.
>
> While we acknowledge that extreme complexity is a field-wide challenge [1], **the above three mechanisms effectively reduce it from an exponential search problem to a tractable engineering task.**
>
> For capability boundary, our neur-symbolic engine is **already covers the complete taxonomy of statically detectable errors defined in the NL2SQL-BUGs benchmark [2] (original Table 1)**. Using NL2SQL-BUGs benchmark, we have established constraints for these errors to assist us in constrained synthesis. Furthermore, our extensible Dialect Specification currently **supports rules for over 29 SQL dialects, demonstrating robust adaptability**.
>
> The capability boundary Line ends exactly where a repair would require a global, non-local refactoring (e.g., deeply nested recursive CTEs with intricate constraint interactions [1]) whose correctness cannot be deterministically verified (future revised in Appendix D.2 ).
>
> **Q3:** When handling empty results, why not continue using the neuro-symbolic collaborative approach (e.g., having the symbolic engine generate candidate repairs and then having the LLM judge the intent)? Why does it instead revert to a purely deterministic, restricted rule?
>
> Yes, we explicitly evaluated this exact neuro-symbolic collaborative approach (labeled **Alignment-LLM in Table 7**), but empirical results compelled us to reject it in favor of the deterministic rule.
> 1. Empirical Evidence: The experiment provided a clear mandate: while Alignment-LLM successfully reduced empty results, it **introduced 21 new semantic errors**, directly violating our framework's **do no harm** principle. In contrast, **our deterministic rule achieved comparable accuracy with zero new errors and was 30x faster**.
> 2. Principled Decision: We chose to revert to the deterministic rule to ensure provable safety. Unlike the probabilistic LLM, **our restricted rule triggers repairs only when there is unambiguous, data-driven evidence for a single correct alternative, ensuring that we never erroneously fix that corrupt valid queries (original lines 335-351).**
>
> **References:**
> 1. SQL Has Problems. We Can Fix Them: Pipe Syntax In SQL (VLDB'24)
> 2. Nl2sql-bugs: A benchmark for detecting vsemantic errors in nl2sql translatio (KDD'25)

---

### Official Review · Reviewer_WeWD · 2025-11-01

**Soundness:** 2
**Presentation:** 2
**Contribution:** 2
**Rating:** 2
**Confidence:** 4

**Summary:**

This paper proposes Post-SQLFix, a neuro-symbolic framework designed for verifiable SQL repair in text-to-SQL tasks. The core idea is to replace ambiguous execution feedback—which often provides low-level, dialect-dependent error information—with a formal synthesis-based repair mechanism. The framework introduces a dialect-independent canonical query structure (CQS) as an intermediate representation, performs layered validation (including syntax, context-free, and context-sensitive semantic checks), and formulates the repair problem as a constraint synthesis task, generating a set of formally valid repair candidates. The LLM then performs constrained generation based on these verified candidates. Experiments on the BIRD and Spider datasets demonstrate an 11.6% improvement in execution accuracy and a 50% reduction in repair iterations compared to EF baselines. The authors also provide proofs of reliability for diagnosis and repair, along with ablation studies and efficiency analysis.

**Strengths:**

1. The paper correctly identifies a major limitation in current text-to-SQL pipelines: the reliance on ambiguous execution feedback that provides low-level, semantically sparse information. Addressing this bottleneck is both timely and significant for the field.

2. The proposed Canonical Query Structure (CQS) and the hierarchical validation layers (α, β, γ) are systematically designed. Formulating the repair synthesis as a constraint satisfaction problem demonstrates formal rigor and represents a well-constructed framework.

**Weaknesses:**

1. The formal verification and repair synthesis rely heavily on explicit schema checking and rule-based validation. The scalability of this approach remains unclear when applied to large or noisy real-world schemas, complex subqueries, or non-standard SQL constructs.

2. While the quantitative accuracy is reported in the tables, the paper would benefit from deeper qualitative examples demonstrating how Post-SQLFix resolves specific semantic errors more effectively than EF or LLM reflection. Concrete before-and-after SQL repair examples would help readers better understand its advantages.

3. The "formal verification + constrained repair" concept closely resembles compiler-inspired static analysis pipelines, with the primary innovation lying in its integration with LLMs—which feels incremental. While the authors extensively formalize the symbolic component, the role of the "neural agent" remains underspecified.

**Questions:**

1. How does the proposed method handle semantically incorrect yet executable queries that return non-empty but erroneous results?

2. How robust is the system to LLM instruction non-compliance? What safeguards exist if the LLM disregards the formal repair constraints and generates a syntactically valid but semantically divergent query?

3. To what extent can this framework be generalized to other code generation tasks beyond text-to-SQL?

---

> ### Author Response · Authors · 2025-11-23
> **Part 1**
>
> We thank the reviewer for **recognizing the timeliness of addressing the execution feedback (EF) bottleneck and for their positive assessment of our systematic CQS design and the formal rigor of our constraint satisfaction framework**. We appreciate your constructive comments regarding the scalability of symbolic analysis on complex real-world schemas (W1), the call for concrete qualitative evidence (W2), and the inquiry into the nature of our neuro-symbolic innovation (W3). We also value the insightful questions regarding safeguards against non-compliance (Q2) and the handling of logical semantic errors (Q1). In this response, we show our neuro-symbolic engine is a **scalable foundation that superior probabilistic models**. Furthermore, we clarify that Post-SQLFix represents a **fundamental architectural** shift—transforming the LLM from an unconstrained predictor into a verifiably constrained executor, and show how our multi-layer **safeguards ensure robustness** even against non-compliant agents.
>
> **Q1**: How does the proposed method handle semantically incorrect yet executable queries that return non-empty but erroneous results?
>
> 1. Post-SQLFix is explicitly designed to guarantee **formal correctness**, not semantically correctness. This is a deliberate scope choice driven by the **principle of "Zero False Positives**. Formal correctness is the non-negotiable prerequisite. A query that cannot be executed (due to syntax or schema errors) is useless, regardless of its intent. **Our goal is to solve this foundational layer perfectly. By replacing unreliable execution feedback with a deterministic engine, we ensure the SQL is valid and runnable.**
> 2. **Architectural Positioning (Plug-and-Play):** Post-SQLFix is designed as a deterministic backend module. Intent Alignment is an orthogonal task best handled by a specialized "Frontend" (e.g., a Reasoning Agent or a separate research module). **Post-SQLFix's Role is to act as the final safety guard. Users can plug Post-SQLFix after any front-end that focuses on specific functions to ensure that its output conforms to the database schema**.
>
> **Q2**: How robust is the system to LLM instruction non-compliance? What safeguards exist if the LLM disregards the formal repair constraints and generates a syntactically valid but semantically divergent query?
>
> Our entire framework is built on a simple but powerful design principle: **we do not *trust* the LLM; we *constrain* and *verify* it.** Our safeguards are based on a deterministic, multi-layered architecture.
>
> **1. Level 1: Constrained Synthesis via Repair Specification ($S_R$).**
> The LLM is not tasked with unconstrained generation. The `Repair Specification ($S_R$)` transforms its role into **constraint-satisfying code synthesis**.
>
> **2. Level 2: Deterministic Action Verification.**
> Our system knows exactly which `fix_action` the LLM was instructed to perform. We can then **deterministically verify** if the output query reflects that specific action. For example, if the action was `REPLACE 'col_A' WITH 'col_B'`, a simple AST-diff or string match on the output can confirm compliance. If the LLM fails to follow this explicit instruction, the output is immediately rejected as non-compliant.
>
> **3. Level 3: The Deterministic Verification Loop as a Failsafe.**
> Every LLM-generated output, $Q'_{str}$, is **always fed back into our complete hierarchical validation engine** (Figure 1b). This loop deterministically catches any non-compliance:
> * If the LLM disregards instructions and generates a query that is still formally incorrect, the loop continues.
> * If it generates a syntactically valid but semantically divergent query (as you suggest), this query will be rejected by validation from AST.
>
> **Empirical Evidence of Robustness.** Our experiments confirm that this constraint-and-verify loop is highly effective. We analyzed instances where different models failed to follow the `$S_R`'s instructions during TA-SQL repair on BIRD:
> | Model| Non-Compliant Errors | Captured |
> |--------------------|:--------------------:|:----------------------:|
> | Qwen2.5-7B   | 2| 100%|
> | Qwen2.5-32B  | 0| N/A|
> | Codestral-7B | 2| 100%|
> | gpt-oss--120B | 15| 100%|
>
> **Q3**: To what extent can this framework be generalized to other code generation tasks beyond text-to-SQL?
> The framework is architecturally universal for any code generation task governed by formal specifications (e.g., Python, Verilog, or OCaml).
> 1. Universal Logic: The core idea is separating "Strict Checking" (Symbolic) from "Creative Writing" (Neural)—works for any programming language. As long as the code can be checked by a compiler or a rule set, our framework applies
> 2. Modular Adaptation: Moving to a new domain is just a matter of swapping the specifications. You simply replace the SQL-specific grammar rules with the target language's syntax tree. The underlying workflow: diagnose, create plan, execute fix, remains exactly the same.

---

> ### Author Response · Authors · 2025-11-23
> **Part 2**
>
> **W1:** The formal verification and repair synthesis rely heavily on explicit schema checking and rule-based validation. The scalability of this approach remains unclear when applied to large or noisy real-world schemas, complex subqueries, or non-standard SQL constructs.
>
> We designed Post-SQLFix with these specific scalability challenges in mind and we have some solution to extend:
> 1. On Large or Noisy Schemas:
> We argue that schema scalability is primarily an upstream challenge (e.g., Schema Linking) rather than a repair-specific bottleneck. As a plug-and-play module, Post-SQLFix inherits the context of the upstream generator.
> 2. On Complex Subqueries, CTEs or Non-Standard SQL Constructs:
> - Recursive Decomposition & Convergence: Our CQS architecture is inherently recursive. It decomposes deeply nested queries (CTEs, subqueries) into independent, atomic scopes for validation. As formalized in our Order-Independent Reachability theorem (detailed in future revised Appendix G and our response to Reviewer XYmR's Q2), this ensures that even the most complex query can be resolved through a finite series of atomic repair steps. **No matter the nesting depth, the system is guaranteed to converge step-by-step to a formally correct state without facing combinatorial bottlenecks.**
> - Dialect Extensibility: For non-standard constructs, our engine avoids monolithic rigidity via the dialect specification. This acts as a configurable compiler interface (currently supporting over 29 dialects via YAML configuration), enabling the system to support dialect-aware functions or operators through simple configuration rather than architectural refactoring.
>
> **W2:** While the quantitative accuracy is reported in the tables, the paper would benefit from deeper qualitative examples demonstrating how Post-SQLFix resolves specific semantic errors more effectively than EF or LLM reflection. Concrete before-and-after SQL repair examples would help readers better understand its advantages.
>
> We have included three detailed qualitative examples in Figures 2, 3, and 4 in the original Appendix that provide exactly this kind of before-and-after comparison. Figure 3 explicitly contrasts our effectiveness with the EF baseline, demonstrating how Post-SQLFix resolves errors in a single deterministic pass while EF struggles with multiple iterations. Figure 4 further showcases the framework's robustness in handling complex scenarios, such as PostgreSQL-specific function constraints and type mismatches. The corresponding prompts are also provided in the Appendix.
>
> **W3:** The "formal verification + constrained repair" concept closely resembles compiler-inspired static analysis pipelines, with the primary innovation lying in its integration with LLMs—which feels incremental. While the authors extensively formalize the symbolic component, the role of the "neural agent" remains underspecified.
>
> We respectfully argue that this resemblance is superficial. Our work does not merely integrate an LLM into a static pipeline; **it fundamentally inverts the compiler paradigm and rigorously redefines the neural agent's role.** We address these two concerns separately below.
> 1. **Paradigm Shift***: The comparison to compiler pipelines captures the components but misses the architectural flow, which is structurally inverse to traditional static analysis. Traditional compilers validate code against a fixed set of rules. They function as checkers that report violations. We does not just check, but also synthesizes dynamic constraints. Our engine compiles abstract diagnostics into a query-specific Repair Specification ($S_R$). The innovation is not simply "adding an LLM to a compiler," but creating a Constraint-Satisfaction Loop. We transform the "Repair" task from an open-ended generation problem (which is fragile) into a constrained synthesis problem, where the symbolic engine acts as the "Specifier" and the LLM acts as the "Realizer." This is a fundamental shift from the standard execution feedback (EF) paradigm.
> 2. On the "Underspecified" Role of the Neural Agent: Regarding the neural agent, we explicitly defined its role in the introduction (original Lines 73-85 and $\S$ 3.5) as "repositioning the LLM from an unreliable arbiter to a constrained neural agent."
> In summary, **far from being incremental, we solves the "Arbitration Paradox"  by stripping the neural agent of the decision-making power it is incompetent to handle, thereby establishing a reliable, formally-grounded repair pipeline**.

---

### Official Review · Reviewer_XYmR · 2025-11-03

**Soundness:** 3
**Presentation:** 3
**Contribution:** 3
**Rating:** 4
**Confidence:** 5

**Summary:**

This paper introduces Post-SQLFix, a neuro-symbolic framework replacing ambiguous Execution Feedback for Text-to-SQL. It formally analyzes queries via a Canonical Query Structure , synthesizes verifiable repair plans , and significantly boosts execution accuracy.

**Strengths:**

S1: This paper addresses the core issue of Text-to-SQL: the "diagnostic poverty" flaw in execution feedback (EF), and the "arbitration paradox" of LLM self-correction. The proposed paradigm shift from "fuzzy feedback" to "verifiable fixes" is highly original and significant, offering a new perspective for building reliable Text-to-SQL systems.

S2: The methodology in the paper is rigorous and clear. Drawing from compiler theory, it constructs a robust neural-symbolic framework. The design of converting SQL into a dialect-agnostic "Canonical Query Structure" (CQS) as an intermediate representation (IR) is an effective approach. On this basis, the design of a hierarchical validation engine (α, β, γ) with a strict partial order (α < β < γ) is logically sound and effectively prevents "cascading false positives." Detailed formal definitions and soundness proofs (Appendix B) greatly enhance the quality of the work.

S3: Another major innovation in this paper is the redefinition of the role of LLMs: from an unreliable "arbiter" to a "constrained agent." Diagnosis and repair plan synthesis are handled by the symbolic engine (providing "what" and "how"), while the LLM is responsible for selecting from the valid plans using its semantic understanding (providing "which"). This neural-symbolic collaboration is outstanding.

S4: Experimental results are convincing. Post-SQLFix significantly improves accuracy on the BIRD and Spider benchmarks (up to +11.6%) and achieves far better repair efficiency (50% fewer iterations) compared to the EF baseline. The multi-dialect analysis (Table 9) is particularly compelling, exposing the vulnerabilities of the EF paradigm in handling errors across different dialects, which contrasts with the robustness of CQS and formalized diagnosis.

**Weaknesses:**

W1: The limitation of the framework is that it mainly addresses "formally incorrect" queries but struggles with "semantically valid but logically erroneous" queries (e.g., incorrect joins, aggregation scope). The paper admits that EF provides "no signal" for such errors. The proposed "empty result alignment" strategy is a narrow heuristic, and does not address a broader range of logical bugs. Although "intent completeness" is undecidable (Appendix B.4.3), the gap between "formally correct" and "intentionally correct" remains significant.

W2: The complexity of building the core "strategy map M" in the "repair synthesis" module seems to be underestimated. The paper should clarify how M is constructed (e.g., is it manually defined based on NL2SQL-BUGs?). The claim that M is a "total function" to ensure "repair completeness" may imply an extremely large and difficult-to-maintain engineering task, particularly when covering all SQL errors and dialects.

W3: The setup of the ablation study (Table 8) is not sufficiently clear. Removing "Plan Synthesis" leads to the largest performance drop (~3%), which suggests that the LLM only receives diagnosis (V_abs) and is forced to perform "speculative" repairs. The meaning of "w/o Hierarchical Validation" is also unclear (does it refer to flattening the validation?). Since the paper emphasizes that hierarchical validation prevents "cascading false positives," providing a specific failure case would make the ablation study more convincing.

W4: CQS is claimed to be "dialect-agnostic," but this is misleading. Validation (especially Layer γ) and repair synthesis are clearly "dialect-aware," relying on the "dialect specification Δ." The paper should clarify the engineering effort required to support a new dialect (e.g., MS SQL Server or Oracle), which is closely related to the maintenance cost of the "strategy map M" mentioned in W2.

**Questions:**

Q1: (Related to W1) Given that the current framework primarily addresses formal errors, what are the plans for handling "semantically valid but logically erroneous" queries (i.e., logical bugs)? The "empty result alignment" strategy is a narrow heuristic—could your formal synthesis framework be extended to diagnose and fix a wider range of logical errors?

Q2: (Related to W2/W4) For queries with multiple complex violations, does the set of repair plans {Λ} generated during "repair synthesis" face combinatorial explosion? How does the framework manage this complexity (e.g., pruning)? Or does the LLM need to handle a large "repair specification" (S_R) containing many candidate plans?

Q3: (Related to Appendix D.2) The residual error analysis attributes failures to the "engineering challenges" of fixing complex SQL (e.g., nested, recursive). Does this specifically refer to the incompleteness of the "strategy map M" for such cases (i.e., unable to synthesize plans), or does it indicate that the symbolic engine generated valid plans, but the LLM failed to successfully "follow instructions" for complex code transformations?

---

> ### Author Response · Authors · 2025-11-23
> **Part 1**
>
> Thank you for **recognizing the paper’s original paradigm shift to verifiable fixes, its rigorous neuro-symbolic methodology, the innovative reconceptualization of LLMs as constrained agents, and its strong empirical improvements in accuracy, efficiency, and cross-dialect robustness**. We appreciate the insightful questions regarding the scope of logical error handling (Q1/W1), the clarity of the ablation study (W3), and the engineering implications of dialect-aware support (W4). Here, we clarify **Post-SQLFix prioritizes formal correctness as the non-negotiable prerequisite for query execution, establishing a reliable foundation upon which broader logical repairs can be built.** We provide detailed responses to each point below.
>
> **Q1**(Related to W1) Given that the current framework primarily addresses formal errors, what are the plans for handling "semantically valid but logically erroneous" queries (i.e., logical bugs)? The "empty result alignment" strategy is a narrow heuristic—could your formal synthesis framework be extended to diagnose and fix a wider range of logical errors?
>
> Post-SQLFix is explicitly designed to guarantee **formal correctness**, not intent correctness(i.e., logical bugs). This is a deliberate scope choice driven by the principle of zero false positives. Formal correctness is the non-negotiable prerequisite. A query that cannot be executed (due to syntax or schema errors) is useless, regardless of its intent. **Our goal is to solve this foundational layer. By replacing unreliable execution feedback with a deterministic engine, we ensure the SQL is valid.**
>
> - **Plug-and-Play:** Post-SQLFix is designed as a deterministic backend module. Intent Alignment is an orthogonal task best handled by a specialized frontend (e.g., a Reasoning Agent or a separate research module). **Post-SQLFix's Role is to act as the final safety guard. Users can plug Post-SQLFix after any front-end that focuses on specific functions** to ensure that its output conforms to the database schema.
> - **The Principle Behind "Empty Result Alignment" is priotise Safety:** We had extended "empty result alignment" strategy, which having the symbolic engine generate candidate repairs and then having the LLM judge the intent, named **Alignment-LLM**. We implemented it and reported its performance in the **original paper inTable 7**. The experiment provided a clear evidence: while Alignment-LLM successfully reduced empty results, **it introduced 21 new semantic errors**, directly violating our framework's ***do no harm*** principle. In contrast, **we achieved comparable accuracy with zero new errors and was 30x faster**.
>
> While empty result alignment strategy is narrow, it is designed to be **provably safe**. It only triggers a repair when there is unambiguous, data-driven evidence for a single, correct alternative, thus preventing the introduction of new errors (original paper line 335-351).
>
> **W3**: The setup of the ablation study (Table 8) is not sufficiently clear. Removing "Plan Synthesis" leads to the largest performance drop (~3%), which suggests that the LLM only receives diagnosis (V_abs) and is forced to perform "speculative" repairs. The meaning of "w/o Hierarchical Validation" is also unclear (does it refer to flattening the validation?). Since the paper emphasizes that hierarchical validation prevents "cascading false positives," providing a specific failure case would make the ablation study more convincing.
>
> - Removing "Plan Synthesis" means asking the LLM to follow the diagnostic feedback alone, without the guidance of explicit, synthesized repair instructions. The "w/o Hierarchical Validation" was achieved by removing our hierarchical validation and replacing it with EF. In original Appendix G , we provided a case study, ***there is no hierarchical validation, the "order" exists in the database schema and is also a keyword in $\alpha$ layer. In this case, if there is no such system, it would be unable to report on how to fix the issue or even completely crash***.
>
> **W4**: CQS is claimed to be "dialect-agnostic," but this is misleading. Validation (especially Layer γ) and repair synthesis are clearly "dialect-aware," relying on the "dialect specification $\Delta$." The paper should clarify the engineering effort required to support a new dialect, which is closely related to the maintenance cost of the "strategy map M" mentioned in W2.
>
> - Yes, describing the entire framework as "dialect-agnostic" was imprecise. We will revise the manuscript to describe the framework as "dialect-aware" to accurately reflect this architectural differences. When handling a new dialect, the system's AST can **parse over 29 dialects and allow users to configure custom dialects**. Although there may be some differences in functions or escape symbols in each dialect, the users can configure the YAML file to achieve codeless addition of rules. We will discuss handling new dialect in the future revised version.

---

> ### Author Response · Authors · 2025-11-23
> **Part 2**
>
> **Q2** (Related to W2/W4) For queries with multiple complex violations, does the set of repair plans {Λ} generated during "repair synthesis" face combinatorial explosion? How does the framework manage this complexity (e.g., pruning)? Or does the LLM need to handle a large "repair specification" (S_R) containing many candidate plans?
>
> Yes, combinatorial explosion is an inherent challenge in program repair, particularly when dealing with multiple, interacting violations [4]. This is **a universal difficulty acknowledged across the field**, as noted in recent work on template-guided repair [2].
> We address the explosion problem by **drastically reducing the search space before it reaches the synthesis stage**. This aligns with findings from CURE [1], which demonstrate that effective repair requires domain-specific constraints to prune low-quality candidates. We implement this through two layers:
> - Hierarchical Pruning: Our validation layers operate sequentially ($\alpha \prec \beta \prec \gamma$). Resolving a single foundational syntactic error often eliminates a cascade of downstream semantic "ghost" errors, preventing the system from wasting resources repairing symptoms rather than root causes.
> - Schema Constraints: Instead of blind enumeration, our neuro-symbolic engine uses database constraints (Foreign Keys, Type Compatibility) to invalidate impossible repairs immediately. **This ensures that the set of generated plans remains compact and relevant.**
>
> The LLM does not handle a large specification containing all possible candidate plans. By the time the Repair Specification ($S_R$) reaches the LLM, the "search" is already finished by the neuro-symbolic engine.
> - The neuro-symbolic engine handles the combinatorial complexity (filtering thousands of possibilities down to a few valid plans).
> - The LLM acts strictly as a constrained executor. It receives only the final, pruned, and verified plans. Its cognitive load is limited to translating these high-level instructions into SQL code, rather than navigating the repair search space itself.
>
>
> **Q3** (Related to Appendix D.2) The residual error analysis attributes failures to the "engineering challenges" of fixing complex SQL (e.g., nested, recursive). Does this specifically refer to the incompleteness of the "strategy map M" for such cases (i.e., unable to synthesize plans), or does it indicate that the neuro-symbolic engine generated valid plans, but the LLM failed to successfully "follow instructions" for complex code transformations?
>
> The "engineering challenges" refer to **both distinct phenomena, representing the logical boundaries of our architecture**. This decoupling of diagnosis from execution is a deliberate architectural feature that allows us to isolate bottlenecks:
>
> - Symbolic analysis limitations: Some SQL constructs involve complex interactions that our current strategy map $M$ cannot analyze while maintaining formal soundness guarantees. Rather than risk introducing new errors through speculative repairs, the system abstains. The Strategy Map $M$ is not a static, hard-coded list. Instead, it is designed as an extensible compiler interface. Current Status: our current implementation **already covers the complete taxonomy of statically detectable errors defined in NL2SQL-BUGs(original Table1)***.
> - Neural instruction limitations: Our scaling analysis (Appendix D.1 and Table 8) demonstrates that instruction-following capabilities improve substantially with model size. This component is being actively addressed by advances in LLM architectures and represents a different engineering trajectory than symbolic coverage expansion.
>
> **The Fundamental Trade-off: Soundness vs. Coverage**: Our decision to abstain in these edge cases reflects a well-documented tension in program analysis. "In Defense of Soundiness" [1] establishes that virtually no realistic analysis tool maintains perfect soundness across all features, as doing so makes analyses "unscalable or imprecise." For SQL, this is amplified by the language's inherent design complexity, as noted in recent industrial research [2].
>
> In conclusion, we prioritize **formal guarantees within a well-defined scope** over universal coverage without assurances. The advantage of our symbolic and neural separation is that **progress on either front extending the symbolic map $M$ or employing stronger LLMs improves system performance without requiring architectural redesign**.
>
> **References:**
> 1. CURE: Code-Aware Neural Machine Translation for Automatic Program Repair (ICSE'21)
> 2. Template-Guided Program Repair in the Era of Large Language Models (ICSE'25)
> 3. In Defense of Soundiness: A Manifesto (Communications of the ACM'15).
> 4. SQL Has Problems. We Can Fix Them: Pipe Syntax In SQL (VLDB'24)

---

### Author Response · Authors · 2025-11-23
**Summary of Common Comments**

We are deeply grateful that the reviewers unanimously recognized Post-SQLFix as a significant advancement in the field. Specifically, Reviewer XYmR, Reviewer 4g3n and Reviewer vH8N **highlighted our work as a highly original** and **powerful paradigm** that successfully transforms the uncertain execution feedback loop into a deterministic diagnose-synthesize-execute workflow, effectively resolving the arbitration paradox. We appreciate Reviewer WeWD for **commending the formal rigor and systematic design of our Canonical Query Structure and hierarchical validation engine**. Furthermore, we value Reviewer vH8N and Reviewer 4g3n's **validation of our "fairly exhaustive" experiments**, particularly noting the framework's superiority in handling dialect-specific errors where traditional methods fail.
Before addressing specific questions, we summarize our responses to three common themes raised across the reviews regarding **Scope**, **Scalability**, and **Generalizability**.

### 1. Our Scope
While reviewers(4g3n, XYmR and WeWD) noted our focus on Formal Correctness (executability) rather than Intent Correctness, we emphasize that this is a deliberate architectural feature designed to **guarantee Zero False Positives** (detailed in original $\S$3.6 and Introduction).
- The Prerequisite: A query that cannot execute (due to schema/syntax errors) is useless. We solve this foundational layer deterministically.
- **The Safety Guarantee:** We prioritize **Do No Harm**. Our system acts as a reliable **backend safety guard**, strictly fixing provable errors while abstaining from speculative changes. This makes Post-SQLFix a robust Plug-and-Play module compatible with any intent focused frontend. We chose to revert to the deterministic rule to ensure provable safety. Unlike the probabilistic LLM, **our restricted rule triggers repairs only when there is unambiguous, data-driven evidence for a single correct alternative, ensuring that we never erroneously fix that corrupt valid queries (original lines 335-351).**

### 2. Innovation & Scalability
Addressing reviewers'(XYmR and 4g3n) questions on our system's capability of managing complex nesting and combinatorial explosion, we clarify that our symbolic engine **resolves the complexity issues inherent in pure neural approaches through architectural inversion and theoretical guarantees**.

- Taming Complexity: PostSQL-Fix handle complexity through **Aggressive Pruning**, using hierarchical and schema constraints (Figure 1(b) and $\S$ 3.2 ) and **Recursive Decomposition**, breaking nested queries into atomic scopes (line 187-192).
- Theoretical Convergence: We can prove that we **do not need to search the entire combinatorial space**. Specifically, finding any valid repair step is sufficient to guarantee convergence to a correct state, rendering the process computationally tractable. We will add the proof, **Theorem 8 (Order-Independent Reachability)**, to the revision.
- Coverage: PostSQL-Fix achieves **full diagnostic completeness over the statically detectable error taxonomy defined in the NL2SQL-BUGs benchmark (original Table 1)**.
- Architectural: PostSQL-Fix do not merely check code like a compiler. We **synthesize dynamic constraints ($S_R$)**. We fundamentally reframe the LLM from an unconstrained *reasoner* to a verifiably constrained *executor*.

### 3. Generalizability: Models and Dialects
Regarding concerns of the framework's generalizability across SQL dialects and integration pipelines (XYmR, WeWD, and 4g3n), we demonstrate that **Post-SQLFix operates as a universal 'Plug-and-Play' backend that enhances reliability regardless of the upstream configuration**.

- **Dialect Aware:** We clarified that our system uses a configurable dialect specification. PostSQL-Fix **supports YAML configuration to handle over 29 dialects or even a new customized dialect**.
- **Plug-and-Play**: We emphasize that Post-SQLFix operates as a backend safety guard. PostSQL-Fix can be **integrated behind any frontend**(e.g., multi-agent systems or intent alignment system) to guarantee formal correctness, effectively decoupling the validity burden from the intent generation.

---

### Author Response · Authors · 2025-11-27
**Summary of Changes in the Revised Paper**

We have revised the manuscript to address the reviewers' constructive feedback, improving theoretical rigor, empirical comprehensiveness, and contextual positioning. Key changes include:

### 1. Theoretical Enhancements (Addressing XYmR-Q2/Q3, WeWD-W1, 4g3n-Q2/W2)
* We refined the Canonical Query Structure (CQS) definition to explicitly model Nested Scopes ($S$) (Section 3.1). We further clarified that the recursive predicate ($P_{recurse}$) in Layer $\gamma$ acts as a trigger to invoke the full validation pipeline for each nested unit. This rigorously demonstrates our framework's structural capability to handle complex nested queries and CTEs **(Addressing WeWD-W1, XYmR-Q3)**.
* We added Theorem 8 (Order-Independent Reachability) in Appendix G. This formally proves that our repair process converges to a correct state without requiring exhaustive search, theoretically resolving the "combinatorial explosion" concern **(Addressing XYmR-Q2, 4g3n-Q2)**.
* We clarified the **Strategic Mapping ($M$)** as a configurable, extensible interface that acts as a safety filter against speculation (Section 3.3.2) **(Addressing XYmR-W4, WeWD-Q2)**.

### 2. Expanded Related Work (Addressing WeWD-W3, vH8N-Q9)
* We expanded Section 2 to include a detailed comparison with state-of-the-art **Neuro-Symbolic Code Generation** methods, specifically **Constrained Decoding** (e.g., Synchromesh). We clarified that unlike generation-time constraints which handle local syntax, Post-SQLFix performs holistic post-hoc diagnosis to address non-local semantic errors **(Addressing WeWD-W3, vH8N-Q9)**.

### 3. New Experiments & Data (Addressing 4g3n-Q1, WeWD-Q3)
* We added **Table 13** (Appendix D.3) presenting results on **four additional LLMs** (Llama-3.1-70B, Mistral-Small, Mamba, GPT-OSS). The results confirm that Post-SQLFix delivers consistent improvements across diverse architectures (Chat & Code models) **(Addressing 4g3n-Q1, WeWD-Q3)**.
* We added **Table 6** to quantify the "Non-Compliance Rate" of LLMs, empirically validating that our validation loop effectively captures potential semantic drifts **(Addressing WeWD-Q2, 4g3n-Q1)**.

### 4. Presentation & Corrections (Addressing 4g3n-W3, vH8N-Q1-Q5)
* We added detailed introductory text to **Appendix F (Table 14)** to better explain the "What-Why-How" logic of our repair paradigms. We have made readability improvements to Formulas 1, 5, 8, and 9 and added more explanations.**(Addressing 4g3n-W3)**.
* We unified function naming conventions (e.g., `SYNTHESIZE_PLANS`, `CREATE_SPECIFICATION`) throughout the main text and appendices to ensure mathematical consistency **(Addressing vH8N-Q3)**.
* We corrected the typo in **Table 5** regarding the Time metric direction and fixed referencing errors regarding the repair mapping table **(Addressing vH8N-Q1/Q5)**. We also corrected all dialect-agnostic to be **dialect-aware** **(Addressing XYmR-W4)**.

---

### Meta-Review · Area_Chair_Ng6T · 2026-01-07

**Summary:**

This work introduces Post-SQLFix, a neuro-symbolic framework that instigates a paradigm shift from ambiguous feedback to verifiable repair, designed to supersede the traditional database execution–feedback paradigm. Experiments on the BIRD and Spider datasets demonstrate an 11.6% improvement in execution accuracy and a 50% reduction in repair iterations compared to execution feedback.

The paper proposes a novel paradigm for Text-to-SQL repair by transforming the uncertain, LLM-guess-dependent “execution feedback” loop into a deterministic and formal “diagnose–synthesize–execute” workflow. During the rebuttal, the authors enhance the theoretical analysis, add new experiments involving additional LLMs, and resolve some presentation issues. However, as most reviewers pointed out, the work focuses solely on “formal correctness,” and its generalizability is limited to SQL. As a result, the current version does not meet the acceptance bar.

**Reviewer Concerns:**

The main concerns with this work are its limited methodological scope focused on formal correctness, its restricted generalizability across SQL dialects, and several presentation issues. Although the authors clarified their intentions, added new experiments involving additional LLMs, and resolved some presentation issues, these improvements only partially address the limitations and thus constrain the overall contribution of the work.

**Reviewer Scores:**

The rebuttal clarifications may warrant a slight score increase, but they are not sufficient to change the acceptance decision.

---

### Decision · Program_Chairs · 2026-01-26

Reject